# Unbalanced Optimal Total Variation Transport: A Theoretical Approach to Spatial Resource Allocation Problems

**Nhan-Phu Chung**[*]
Institute of Applied Mathematics
University of Economics Ho Chi Minh City
phucn@ueh.edu.vn

**Jinhui Han**[*]
Guanghua School of Management
Peking University
jinhui.han@gsm.pku.edu.cn

**Bohan Li**[*]
Center for Financial Engineering
Soochow University
bhli@suda.edu.cn

**Zehao Li**[*†]
Guanghua School of Management
Peking University
zehaoli@stu.pku.edu.cn

## Abstract

We propose and analyze a new class of unbalanced weak optimal transport (OT) problems with total variation penalties, motivated by spatial resource allocation tasks. Unlike classical OT, our framework accommodates general unbalanced non-negative measures and incorporates cost objectives that directly capture operational trade-offs between transport cost and supply–demand mismatch. In the general setting, we establish the existence of optimal solutions and a dual formulation. We then focus on the semi-discrete setting, where one measure is discrete and the other is absolutely continuous, a structure relevant to applications such as service area partitioning for facilities like schools or medical stations. Exploiting a tessellation-based structure, we derive the corresponding explicit optimality conditions. We further address a quantization problem that jointly optimizes the locations and weights of discrete support points, applicable to facility location tasks such as the cost-efficient deployment of battery swap stations or e-commerce warehouses, informed by demand-side data. The dual-tessellation structure also yields explicit gradient expressions, enabling efficient numerical optimization in finite dimensions.

## 1 Introduction

Optimal Transport (OT) provides a principled way to compare and transform probability measures by reallocating mass at minimal cost, as determined by a ground metric. Its sensitivity to distributional geometry and capacity to interpolate between measures have made OT a central tool in machine learning, statistics, and applied mathematics [51]. Applications span generative modeling [6, 18, 21], domain adaptation [10, 16, 23, 28], robust statistics [45, 46], clustering [35], image and shape processing [49, 59], graph matching [29], stochastic dynamics [66], and posterior inference [30].

Classical OT enforces strict mass conservation, which is often impractical in real-world scenarios due to noise, outliers, or domain mismatch [45, 58]. Unbalanced OT (UOT) addresses this limitation by relaxing marginal constraints through penalty functions [42, 43]. Various penalties have been

---

[*]Equal-contribution authorship listed in alphabetical order.
[†]Corresponding author.

39th Conference on Neural Information Processing Systems (NeurIPS 2025).

introduced to accommodate different applications, such as Kullback–Leibler (KL) divergence [20, 43], $f$-divergences [43], $\chi^2$-divergence [10], and total variation (TV) [45, 53, 54, 62]. Correspondingly, a range of computational strategies have been developed, including regression-based approaches [17], generalized Sinkhorn algorithms [20, 52], low-rank solvers [57], and GAN-based methods [65]. Despite these advances, the theoretical foundations of UOT under non-superlinear penalties, particularly TV, remain underdeveloped.

Meanwhile, the semi-discrete OT setting, where one marginal is continuous and the other discrete, has gained prominence in applications like autoencoders [5, 63] and GAN training [4]. Semi-discrete UOT further integrates this structure with mass imbalance, and has proven effective in tasks like quantization [15] and mode regularization in GANs [64]. In parallel, weak OT formulations generalize classical couplings by evaluating cost over disintegration of transport plans, allowing each source point to be matched with a probability distribution over the target space [2, 7, 19, 22]. This added flexibility enhances robustness to sampling noise and is particularly advantageous in settings where empirical distributions render classical couplings unstable [3, 8, 29].

In this paper, motivated by spatial resource allocation problems, we study a new class of unbalanced weak OT problems with total variation penalties. Unlike smoother, superlinear divergences studied in existing theory, TV-based penalties are piecewise linear and less smooth, posing substantial analytical challenges. Our work advances the theory of OT by developing a rigorous framework for this more general and practically motivated setting.

**Our Contributions.**

- **A novel UOT formulation.** We propose the first formulation of *semi-discrete, unbalanced weak optimal transport with total variation penalties*, integrating weak couplings, mass imbalance, and non-superlinear regularization. This formulation is motivated by practical applications and introduces new mathematical structures.
- **Comprehensive theoretical results for TV-penalized UOT.** We establish the existence of optimal solutions and derive a dual formulation in the general setting. In the semi-discrete case, we develop a tessellation-based reformulation that reduces the infinite-dimensional problem to a finite-dimensional one, enabling explicit optimality conditions via Laguerre cell partitioning. Building on this, we further address an associated quantization problem. These results significantly extend and generalize classical entropy-regularized transport frameworks to the non-smooth, TV-penalized regime.
- **Practical significance.** Our theoretical framework captures key features of real-world problems, including semi-discrete geometry, mass imbalance, TV-type regularization, and probabilistic routing. In the semi-discrete regime, the induced dual-tessellation structure and explicit gradient expressions enable efficient optimization, which we demonstrate through two illustrative applications: *supply area division* and *spatial resource allocation*.

**Related Work.**

From a theoretical standpoint, recent work has significantly advanced the understanding of OT under classical assumptions and various extensions [12, 25, 26, 31, 36, 44, 47, 56]. Among these, Pooladian et al. [55] investigate the semi-discrete OT problem with entropic regularization, followed by Agarwal et al. [1], which develops a combinatorial algorithm for the same setting. Theoretical analysis of UOT has also emerged, although much of the focus has been on algorithmic convergence and scalability rather than fundamental properties of the UOT problem itself [14, 24, 52, 60, 66]. A notable exception is the work of Bourne et al. [15], which addresses semi-discrete UOT with smooth entropy-based penalties. Besides, weak OT has received growing attention [9, 38]. Chung and Trinh [22] develop a general duality theory for weak optimal entropy transport under superlinear convex penalties. Most recently, Beiglböck et al. [11] extend strong duality results to weak OT with nonlinear cost functions, establishing a fundamental theorem for weak OT.

## 2 An unbalanced weak optimal total variation transport problem

### 2.1 Background on optimal transport and its generalizations

Rooted in the seminal work of Monge and Kantorovich, the classical OT problem seeks the most cost-efficient plan to transport mass from a source distribution $\mu_1$ on $X_1$ to a target distribution $\mu_2$ on

$X_2$ under a prescribed cost function $c(x_1, x_2)$. Formally, it solves

$$\min_{\pi \in \Pi(\mu_1, \mu_2)} \int_{X_1 \times X_2} c(x_1, x_2) d\pi(x_1, x_2),$$

where $\Pi(\mu_1, \mu_2)$ denotes the set of couplings with marginals $\mu_1$ and $\mu_2$. To better accommodate various practical needs, several generalizations of OT have been proposed. Two important directions include *weak OT*, which relaxes the requirement of exact target marginal matching, and *entropy-regularized UOT*, which allows for mass variation while promoting smoother solutions. These extensions have proven particularly valuable in real-world applications, where strict mass conservation or exact marginal fidelity is either infeasible or undesirable.

**Weak OT.** The weak OT framework replaces the hard constraint of matching the target distribution exactly with a more relaxed, aggregate matching condition, typically through expectations or barycentric projections. A representative formulation is:

$$\min_{\pi \in \Pi(\mu_1, \mu_2)} \int_{X_1} C(x_1, \pi_{x_1}) \mu_1(dx_1),$$

where $(\pi_{x_1})_{x_1 \in X_1}$ denotes the disintegration of the coupling $\pi$ with respect to its first margin. Here, the cost function changes to $C : X_1 \times \mathcal{P}(X_2) \to \mathbb{R} \cup \{+\infty\}$, where $\mathcal{P}(X)$ represents the space of probability measures on $X$. Notably, the classical OT formulation is recovered as a special case when $C(x, p) := \int c(x, y) dp(y)$ for a base point-to-point cost $c$ [2, 8, 32]. Weak OT approach proves particularly useful when the target measure $\mu_2$ is only partially observed or known with uncertainty, a common situation in learning systems built on finite data. Compared to the rigid structure of classical OT, the weak formulation enhances stability and robustness in the learned transport plan. Furthermore, with suitable convexity properties, weak OT problems remain computationally tractable and well-suited for integration into learning pipelines.

**Entropy-Regularized UOT.** Another prominent extension is entropy-regularized UOT, which combines the benefits of entropic smoothing with the ability to handle measures of *unequal* total mass. This variant modifies the classical formulation by relaxing marginal constraints and penalizing deviations using divergence terms. A typical formulation, following [43], is

$$\inf_{\gamma \in \mathcal{M}(X_1 \times X_2)} \sum_{i=1}^{2} \mathcal{F}_i(\gamma_i | \mu_i) + \int_{X_1 \times X_2} c(x_1, x_2) d\gamma(x_1, x_2),$$

where $\mathcal{M}(X)$ denotes the space of nonnegative finite Borel measures, and $\gamma_i$ is the margin of the transport plan $\gamma$ on $X_i$. In contrast to classical OT, the plan $\gamma$ is not constrained to be a coupling, and the input measures $\mu_1$ and $\mu_2$ need not have equal total mass. The functions $\mathcal{F}_i$ typically encode divergences such as the KL divergence, thereby penalizing deviation from the prescribed marginals. Intuitively, although perfect mass preservation is inherently unattainable in the unbalanced setting, the resulting formulation aims to strike a balance between minimizing transport cost and permitting controlled deviations from marginal consistency. The entropic terms enhance smoothness and yield strongly convex objectives, enabling fast convergence of iterative solvers such as generalized Sinkhorn algorithms [20, 52]. Notably, the classical OT problem is recovered as a limiting case when both input measures are probability distributions and the penalization for any deviation is infinite.

## 2.2 Our considered problem

In many decision-centric tasks, particularly those arising in operational and resource-constrained environments, the cost of deviating from a desired distribution cannot be adequately captured by standard divergence-based penalties. Although statistical divergences such as the KL divergence are inherently asymmetric, their asymmetry does not generally reflect the directional economic or operational significance observed in real-world applications. For example, under-supplying a resource (e.g., understocking inventory or under-allocating service capacity) typically incurs substantially higher costs than over-supplying, due to lost sales, service failures, or unmet demand. More importantly, classical divergence measures quantify statistical dissimilarity without conveying explicit physical meaning or cost implications, limiting their interpretability in applied settings such as supply-demand matching and resource allocation.

To address this gap, we propose an unbalanced weak optimal total variation transport problem that replaces statistical divergence penalties with a newsvendor-type loss function, a cost model well-established in inventory management and economics [34, 37, 50]. The newsvendor loss offers a

parsimonious yet expressive way of modeling asymmetric penalties by assigning distinct weights to overestimation and underestimation. Its piecewise linear form directly reflects the real-world mismatch level, providing a physically meaningful and interpretable objective that aligns with the goals of many resource allocation problems [33, 67]. We consider the following optimization problem: for nonnegative vectors $\mathbf{k}_1 = (a_1, b_1)$ and $\mathbf{k}_2 = (a_2, b_2)$, $\mu_1 \in \mathcal{M}(X_1)$, $\mu_2 \in \mathcal{M}(X_2)$, and $\gamma \in \mathcal{M}(X_1 \times X_2)$,

$$\mathcal{E}_C^{\mathbf{k}_1, \mathbf{k}_2}(\mu_1, \mu_2) := \inf_{\gamma \in \mathcal{M}(X_1 \times X_2)} \mathrm{ET}_C^{\mathbf{k}_1, \mathbf{k}_2}(\gamma | \mu_1, \mu_2), \tag{1}$$

where

$$\mathrm{ET}_C^{\mathbf{k}_1, \mathbf{k}_2}(\gamma | \mu_1, \mu_2) := a_1 |(\mu_1 - \gamma_1)^+| + b_1 |(\gamma_1 - \mu_1)^+| + a_2 |(\mu_2 - \gamma_2)^+| + b_2 |(\gamma_2 - \mu_2)^+|$$
$$+ \int_{X_1} C(x_1, \gamma_{x_1}) d\gamma_1(x_1),$$

with $\gamma_1$ and $\gamma_2$ denoting the marginals of $\gamma$, and $(\gamma_{x_1})_{x_1 \in X_1}$ being the disintegration of $\gamma$ with respect to its first marginal. Here, for a scalar or measurable function $g$, we set $g^+ := \max\{g, 0\}$, while for a signed measure $\nu$, we take $\nu^+$ to be the positive variation in the Jordan decomposition $\nu := \nu^+ + \nu^-$ with $\nu^+ \geq 0, \nu^- \geq 0$, and $\nu^+ \perp \nu^-$. When interpreted through the lens of supply-demand matching, the objective function offers a clear economic interpretation. The first two terms penalize supply-side mismatches: shortages are penalized at rate $b_1$, while excess supply incurs cost $a_1$. The next two terms represent demand-side penalties: lost demand due to undersupply is penalized at rate $a_2$, whereas overprovision relative to demand incurs cost $b_2$. When all weights are set to one, the objective corresponds to the TV distance between marginals and has been applied in prior work, such as [68].

Following the introduction in Section 2.1, the considered weak formulation can reduce to a normal case by choosing $C(x_1, p) = k \int_{X_2} c(x_1, x_2) dp(x_2)$ for some point-to-point function $c : X_1 \times X_2 \to (-\infty, +\infty]$. This choice leads to a special case of entropy-regularized UOT studied in [43]:

$$\mathcal{E}^{\mathbf{k}_1, \mathbf{k}_2, k}(\mu_1, \mu_2) := \inf_{\gamma \in \mathcal{M}(X_1 \times X_2)} \mathrm{ET}^{\mathbf{k}_1, \mathbf{k}_2, k}(\gamma | \mu_1, \mu_2), \tag{2}$$

where

$$\mathrm{ET}^{\mathbf{k}_1, \mathbf{k}_2, k}(\gamma | \mu_1, \mu_2) := a_1 |(\mu_1 - \gamma_1)^+| + b_1 |(\gamma_1 - \mu_1)^+| + a_2 |(\mu_2 - \gamma_2)^+| + b_2 |(\gamma_2 - \mu_2)^+|$$
$$+ k \int_{X_1 \times X_2} c(x_1, x_2) d\gamma(x_1, x_2),$$

where the nonnegative constant $k$ serves to rescale the transportation cost, reflecting the relative importance between transport and mismatch costs. While technically it can be absorbed into the definition of the cost function $c$, keeping it explicit highlights the application-dependent trade-off.

**Definition 2.1.** *We say that the problem* (1) *(resp.* (2)*) is feasible if there exists a transport plan* $\gamma \in \mathcal{M}(X_1 \times X_2)$ *such that* $\mathrm{ET}_C^{\mathbf{k}_1, \mathbf{k}_2}(\gamma | \mu_1, \mu_2) < \infty$ *(resp.* $\mathrm{ET}^{\mathbf{k}_1, \mathbf{k}_2, k}(\gamma | \mu_1, \mu_2) < \infty$).

While our primary focus application is on resource allocation, we anticipate that the proposed unbalanced weak OT framework holds significant potential across a broader spectrum of applications. As discussed earlier, this setting represents one of the most expressive and versatile generalizations in OT theory. In particular, the weak formulation enables us to push the theory to its limits and includes classical OT as a special case. Beyond its generality, the weak formulation has shown promising practical advantages: by relaxing the coupling constraints, it enhances robustness to sampling noise and stability in empirical settings where exact couplings may not exist or may be sensitive to perturbations (see, e.g., [8, 29]). In this work, our objective is to establish a rigorous theoretical foundation for this framework, paving the way for its application in diverse decision-making and learning contexts.

**Example 2.1** (**Order dispatching in ride-sourcing platforms**). *[68] considers an order dispatching problem in ride-sourcing platforms, where the objective is to assign idle vehicles to meet customer demand across different locations. To simultaneously account for transportation cost and supply-demand mismatches, both of which directly affect platform efficiency and profitability, the authors propose a formulation that can be viewed as a special instance of the model in* (2)*. In their context, problem-specific constraints are imposed to reflect practical considerations, such as a fixed supply-side marginal to represent the limited number of available drivers.*

**Remark 2.1.** *Recasting resource allocation problems as an unbalanced weak OT framework provides several theoretical and practical advantages. First, the OT formulation is geometrically expressive. By embedding the problem in a metric space, it naturally captures spatial relationships through structures such as Laguerre and Voronoi tessellations—features that are difficult to obtain from purely algebraic linear programming (LP) models. This geometric perspective is not only mathematically elegant but also supports practical applications, such as service-area partitioning for facilities like fire stations or warehouses. Second, OT allows a natural treatment of imbalance through TV penalties, which directly quantify the cost of mass creation or deletion. This yields an explicit trade-off between transportation and mismatch costs, avoiding artificial slack variables often used in LP formulations. The magnitude of the TV term provides an intuitive measure of over- or under-supply. Third, the continuous formulation greatly enhances scalability, particularly in semi-discrete formulation, as it avoids high-dimensional LP formulations by reducing the optimization to a weight vector whose size matches the number of facilities. Overall, the OT-based approach offers a geometrically faithful, imbalance-aware, and efficient modeling framework. While LP methods remain convenient for direct implementation, we believe the structure, interpretability, and flexibility of the OT perspective provide substantial benefits.*

## 3   Main theoretical results

In this section, we present our main theoretical contributions, which are organized into three key components. First, in Section 3.1, we derive a dual formulation for the unbalanced weak optimal total variation transport problem (1). This dual characterization serves as a foundational tool for establishing optimality conditions and facilitates both analysis and computation. We then turn our attention to the semi-discrete setting in Sections 3.2 and 3.3, where one marginal is a discrete measure and the other is absolutely continuous with respect to the Lebesgue measure. This setting is motivated by practical applications in spatial resource allocation, including (1) service area partitioning, where one seeks to determine the optimal service regions of facilities such as medical stations or schools, and (2) facility location problems, such as the placement of battery swap stations or e-commerce warehouses. Specifically, Section 3.2 establishes optimality conditions for the semi-discrete transport problem, while Section 3.3 addresses a quantization problem in which both the weights and locations of the discrete marginal are jointly optimized to minimize the total transport cost.

### 3.1   A dual formulation of weak optimal total variation transport

We first introduce some technical notations. For a metric space $X$, we denote by $C_b(X)$ and $B(X)$ the space of bounded continuous real functions and Borel real functions on $X$, respectively. A metric space $X$ is *Polish* if it is complete and separable.

**Definition 3.1.** *We say that a measurable map $C : X_1 \times \mathcal{P}(X_2) \to (-\infty, +\infty]$ has property (T) if for every $M > 0$ and every sequence $\{\gamma^n\}_{n \in \mathbb{N}} \subset \mathcal{M}(X_1 \times X_2)$ such that $\int_{X_1} C(x_1, \gamma^n_{x_1}) d\gamma^n_1(x_1) \leq M$ for all $n \in \mathbb{N}$, the sequence $\{\gamma^n\}_{n \in \mathbb{N}}$ is tight.*

Let $X_1, X_2$ be Polish metric spaces, and let $C : X_1 \times \mathcal{P}(X_2) \to (-\infty, +\infty]$ be a lower semicontinuous function that is bounded from below and satisfies property (T) as defined in Definition 3.1. These assumptions on $C$ are nearly *minimal* and encompass commonly used cost functions as special cases. Assume further that $(b_1 + b_2)/2 + \inf C > 0$. Before proceeding to duality and optimality conditions, it is essential to establish that the unbalanced weak OT problem (1) admits a solution. The following theorem guarantees the existence of an optimal transport plan under these mild conditions.

**Theorem 3.1 (Existence of optimal solutions).** *Let $\mu_i \in \mathcal{M}(X_i)$, for $i = 1, 2$ and suppose that problem (1) is feasible. Then the problem (1) has at least one optimal plan.*

We now proceed to derive the dual formulation of the primal problem (1), which, as previously noted, constitutes a critical intermediate step in our theoretical development. To elucidate the geometric role of the TV penalty in the weak formulation, we begin by introducing a family of auxiliary functions $I_i$ and $J_i$ for $i = 1, 2$, along with the corresponding admissible function classes $\Phi_I$ and $\Phi_J$. These constructions will be used in the strong duality result presented in Theorem 3.2.

For each $i = 1, 2$, we define the functions $I_i : \mathbb{R} \to (-\infty, +\infty]$ by

$$I_1(\varphi) := \inf_{s \geq 0} (s\varphi + a_1(1-s)^+ + b_1(s-1)^+) = a_1 \mathbf{1}_{\{\varphi > a_1\}} + \varphi \mathbf{1}_{\{-b_1 \leq \varphi \leq a_1\}} - \infty \mathbf{1}_{\{\varphi < -b_1\}},$$

$$I_2(\varphi) := \inf_{s \geq 0} (s\varphi + a_2(1-s)^+ + b_2(s-1)^+) = a_2 \mathbf{1}_{\{\varphi > a_2\}} + \varphi \mathbf{1}_{\{-b_2 \leq \varphi \leq a_2\}} - \infty \mathbf{1}_{\{\varphi < -b_2\}}.$$

$$(3)$$

Likewise, we define the conjugate-type functions $J_i : \mathbb{R} \to (-\infty, +\infty]$ by

$$J_1(\phi) = \sup_{s > 0} \frac{\phi - a_1(1-s)^+ - b_1(s-1)^+}{s} = +\infty \mathbf{1}_{\{\phi > a_1\}} + \phi \mathbf{1}_{\{-b_1 \leq \phi \leq a_1\}} - b_1 \mathbf{1}_{\{\phi < -b_1\}},$$

$$J_2(\phi) = \sup_{s > 0} \frac{\phi - a_2(1-s)^+ - b_2(s-1)^+}{s} = +\infty \mathbf{1}_{\{\phi > a_2\}} + \phi \mathbf{1}_{\{-b_2 \leq \phi \leq a_2\}} - b_2 \mathbf{1}_{\{\phi < -b_2\}}.$$

Using these functions, we define two sets of admissible test functions:

$$\Phi_I := \{(\varphi_1, \varphi_2) \in C_b(X_1) \times C_b(X_2) : -b_1 \leq \varphi_1(x_1), -b_2 \leq \varphi_2(x_2) \text{ for every } x_i \in X_i, i = 1, 2$$
$$\text{and } \varphi_1(x_1) + p(\varphi_2) \leq C(x_1, p) \text{ for every } x_1 \in X_1, p \in \mathcal{P}(X_2)\},$$

$$\Phi_J := \{(\varphi_1, \varphi_2) \in C_b(X_1) \times C_b(X_2) : \varphi_1(x_1) \leq a_1, \varphi_2(x_2) \leq a_2 \text{ for every } x_i \in X_i, i = 1, 2$$
$$\text{and } J_1(\varphi_1(x_1)) + p(J_2(\varphi_2)) \leq C(x_1, p) \text{ for every } x_1 \in X_1, p \in \mathcal{P}(X_2)\}.$$

**Theorem 3.2 (Dual representation).** *Suppose that for every $x_1 \in X_1$, $C(x_1, \cdot) : \mathcal{P}(X_2) \to (-\infty, +\infty]$ is convex. Then for every $\mu_i \in \mathcal{M}(X_i), i = 1, 2$, we have*

$$\mathcal{E}_C^{\mathbf{k}_1, \mathbf{k}_2}(\mu_1, \mu_2) = \sup_{(\varphi_1, \varphi_2) \in \Phi_J} \sum_{i=1}^2 \int_{X_i} \varphi_i d\mu_i = \sup_{(\varphi_1, \varphi_2) \in \Phi_I} \sum_{i=1}^2 \int_{X_i} I_i(\varphi_i(x_i)) d\mu_i(x_i).$$

**Remark 3.1.** *A similar dual representation has been established in the case where both $X_1$ and $X_2$ are compact [22, Theorem 2]. Extending this result to general (non-compact) spaces, however, is nontrivial, primarily due to the presence of non-superlinear, newsvendor-type penalties, which are motivated by practical considerations. In this work, we adopt a different proof strategy from that used in [22, Theorem 2] to accommodate these challenges.*

Such a dual perspective plays a central role in the modern development and application of OT theory, owing to both its theoretical and practical advantages. On the theoretical side, the dual formulation provides essential structural insights, such as the characterization of optimal plans through Kantorovich potentials $(\varphi_1, \varphi_2)$ and the derivation of necessary and sufficient conditions for optimality. On the practical side, it often leads to more computationally tractable formulations. Notably, the dual representation established in Theorem 3.2 is very general, and it partially recovers [43, Corollary 4.12] when $C(x_1, p) = k \int c(x_1, x_2) dp$ and the entropy functions are given by $F_i = a_i(1-s)^+ + b_i(s-1)^+$, $i = 1, 2$, as stated in the following corollary.

**Corollary 3.1.** *Let $c : X_1 \times X_2 \to (-\infty, +\infty]$ be a lower semicontinuous function that is bounded from below and has compact sublevels. Suppose that $(b_1 + b_2)/2 + k \inf c > 0$. Then, for every $\mu_i \in \mathcal{M}(X_i), i = 1, 2$, we have*

$$\mathcal{E}^{\mathbf{k}_1, \mathbf{k}_2 \cdot k}(\mu_1, \mu_2) = \sup_{(\varphi_1, \varphi_2) \in \Lambda_J} \sum_{i=1}^2 \int_{X_i} \varphi_i d\mu_i = \sup_{(\varphi_1, \varphi_2) \in \Lambda_I} \sum_{i=1}^2 \int_{X_i} I_i(\varphi_i(x_i)) d\mu_i(x_i),$$

*where the admissible sets $\Lambda_J$ and $\Lambda_I$ are respectively defined as $\Lambda_J := \{(\varphi_1, \varphi_2) \in C_b(X_1) \times C_b(X_2) : \varphi_1(x_1) \leq a_1, \varphi_2(x_2) \leq a_2, J_1(\varphi_1(x_1)) + J_2(\varphi_2(x_2)) \leq k \cdot c(x_1, x_2) \text{ for every } x_1 \in X_1, x_2 \in X_2\}$ and $\Lambda_I := \{(\varphi_1, \varphi_2) \in C_b(X_1) \times C_b(X_2) : \varphi_1(x_1) \geq -b_1, \varphi_2(x_2) \geq -b_2, \varphi_1(x_1) + \varphi_2(x_2) \leq k \cdot c(x_1, x_2) \text{ for every } x_1 \in X_1, x_2 \in X_2\}.$*

In what follows, rather than deriving the optimality conditions directly under the general framework of Theorem 3.2, which is straightforward but offers limited intuitive insight, we shift our focus to more practically relevant semi-discrete settings. This allows for a clearer theoretical development and facilitates meaningful applications, particularly in the context of spatial resource allocation.

## 3.2 Semi-discrete unbalanced total variation transport

Here, we consider a semi-discrete scenario under the problem formulation (2), where the first marginal is continuous and the second marginal is discrete. Specifically, we let $\mu_1 \in \mathcal{M}(X_1)$ be absolutely continuous with respect to the Lebesgue measure, and define $\mu_2 = \sum_{i=1}^{M} m_i \delta_{y_i} \in \mathcal{M}(X_2)$ for some given integer $M > 0$, where $\delta_{y_i}$ denotes the Dirac measure centered at $y_i$. Examples arising in spatial resource allocation, such as the placement of e-commerce warehouses or public facilities like schools, naturally fit within this framework. In such settings, the discrete points represent service stations (i.e., supply locations), while the continuous measure models the spatial distribution of service demand across the region. We first introduce several new notations or concepts that will be used later. For any $a, b \in \mathbb{R}$, we denote $a \vee b := \max\{a, b\}$. Let $n > 0$ be any given integer, and let $X_1$ and $X_2$ be subsets of $\mathbb{R}^n$. Assume that $(b_1 + b_2)/2 + k \inf c > 0$ and $c$ has compact sublevels. In both Sections 3.2 and 3.3, we further assume that $c$ is *radial* according to the following definition (see also [15]).

**Definition 3.2.** *A function $c : X_1 \times X_2 \to (-\infty, +\infty]$ is* radial *if it can be written as $c(x, y) = g(d(x, y))$ with a continuous, strictly increasing function $g : [0, +\infty] \to [0, +\infty]$ satisfying $g(0) = 0$.*

Beyond its practical significance, the semi-discrete setting is of theoretical interest due to the special geometric structure of its solutions. In particular, the optimal transport plan induces a cell decomposition of the domain, which is both mathematically elegant and operationally interpretable.

**Definition 3.3 (Generalized Laguerre cells, [15]).** *Given a transportation cost $c : X_1 \times X_2 \to (-\infty, +\infty]$ and $y_1, \ldots, y_M \in X_2$, we define the generalized Laguerre cells corresponding to the weight vector $w \in \mathbb{R}^M$ as follows: for $i \in \{1, \ldots, M\}$*

$$C_i(w) = \{x \in X_1 \mid c(x, y_i) < +\infty, \, c(x, y_i) - w_i \leq c(x, y_j) - w_j \text{ for all } j \in \{1, \ldots, M\}\}. \quad (4)$$

*The residual set, denoted by $R$, is defined as $R = \{x \in X_1 \mid c(x, y_i) = +\infty\}$.*

The next theorem recasts the dual functional from Theorem 3.2 as a finite-dimensional optimization over the weight vector $w$ associated with the cell division, thereby reducing the original infinite-dimensional problem over Kantorovich potentials to a more tractable finite-dimensional setting.

**Theorem 3.3 (Dual tessellation formulation).** *Suppose that problem (2) is feasible. Then*

$$\mathcal{E}^{\mathbf{k}_1, \mathbf{k}_2, k}(\mu_1, \mu_2) = \sup \left\{ \mathcal{G}(w) \mid w = (w_1, \ldots, w_M) \in [-b_2, +\infty)^M \right\}, \quad (5)$$

*where*

$$\mathcal{G}(w) = \sum_{i=1}^{M} \int_{C_i(w)} I_1(-b_1 \vee [k \cdot c(x, y_i) - w_i]) d\mu_1(x) + a_1 \mu_1(R) + \sum_{i=1}^{M} I_2(w_i) m_i$$

$$= \int_{\Omega \setminus R} I_1(c^w(x)) d\mu_1(x) + a_1 \mu_1(R) + \sum_{i=1}^{M} I_2(w_i) m_i \quad (6)$$

*with $c^w : X_1 \to (-\infty, +\infty]$ defined by $c^w(x) := -b_1 \vee \min_{i=1, \ldots, M} \{k \cdot c(x, y_i) - w_i\}$.*

Equation (6) reveals that the reformulated dual objective $\mathcal{G}$ admits an explicit and tractable dependence on the weight vector $w$, facilitating algorithm design for identifying the optimal weights. Building on Theorem 3.3, we now proceed to establish the corresponding optimality condition.

**Theorem 3.4 (Optimality conditions).** *Suppose that $X_2 = \{y_1, \ldots, y_M\}$ and problem (2) is feasible. Let $\boldsymbol{\gamma} \in \mathcal{M}(X_1 \times X_2)$ be an optimizer for problem (2) and $w = (w_1, \ldots, w_M) \in [-b_2, +\infty)^M$ be an optimizer for problem (5). Then $\boldsymbol{\gamma}$ admits the form $\boldsymbol{\gamma} = \sum_{i=1}^{M} \eta_i \otimes \delta_{y_i}$ for $\eta_i \in \mathcal{M}(X_1), i = 1, \ldots, M$. The Lebesgue decomposition of $\gamma_1$ with respect to $\mu_1$ is $\gamma_1 = \sum_{i=1}^{M} \eta_i = \frac{d\gamma_1}{d\mu_1} \mu_1 + \gamma_1^{\perp}$. Define $S_1 := \{i \in \{1, \ldots, M\} : m_i \neq 0\}$ and $S_2 := \{i \in \{1, \ldots, M\} : m_i = 0\}$. Then the following conditions hold:*

$$\eta_i = \gamma_1 \llcorner C_i(w) \text{ for every } i = 1, \ldots, M \text{ (i.e., } \boldsymbol{\gamma} = \sum_{i=1}^{M} \gamma_1 \llcorner C_i(w) \otimes \delta_{y_i}), \quad (7)$$

*where $\gamma_1 \llcorner C_i(w)$ denotes the restriction of $\gamma_1$ to the cell $C_i(w)$, and*

$$\frac{d\gamma_1}{d\mu_1}(x) = 1 \text{ for } \mu_1\text{-a.e. } x \in X_1 \setminus R, \qquad \frac{d\gamma_1}{d\mu_1}(x) = 0 \text{ for } \mu_1\text{-a.e. } x \in R, \quad (8)$$

$$|\eta_i| = m_i \text{ for all } i \in S_1, \quad c^w(x) = -b_1 \text{ for } \gamma_1^\perp\text{-a.e. } x \in X_1 \setminus R, \tag{9}$$

$$w_i = -b_2 \text{ for every } i \in S_2. \tag{10}$$

*Conversely, if $\gamma$ and $w$ satisfy (7-10), then they are optimal solutions to (2) and (5), respectively.*

The first marginal of the optimal transport plan comprises two components: an absolutely continuous part with respect to the input marginal $\mu_1$, and a singular part that arises due to the piecewise-linear TV penalty, which differs from [15]. In particular, the optimality conditions established in Theorem 3.4 enable efficient recovery of the optimal transport plan once the optimal weight vector $w^*$ has been determined. The latter can be computed using gradient-based methods, as the exact (sub-)gradient of the dual objective $G(w)$ with respect to each $w_i$ is explicitly available and presented below.

**Proposition 3.1** (**Gradient of dual tessellation formulation**). *$\mathcal{G}$ defined in (6) from Theorem 3.3 is (sub-)differentiable with respect to each $w_i$ and*

$$\partial_{w_i}\mathcal{G}(w) = \begin{cases} -\mu_1(C_i(w) \cap \{x : k \cdot c(x, y_i) - w_i \in [-b_1, a_1]\}) + m_i, & w_i \in (-b_2, a_2), \\ -\mu_1(C_i(w) \cap \{x : k \cdot c(x, y_i) - w_i \in [-b_1, a_1]\}) + [0, m_i], & w_i \in \{a_2\}, \\ -\mu_1(C_i(w) \cap \{x : k \cdot c(x, y_i) - w_i \in [-b_1, a_1]\}), & w_i \in (a_2, +\infty). \end{cases}$$

**Remark 3.2.** *It is clear that $\mathcal{G}(w)$ is decreasing in $w_i$ over $(a_2, +\infty)$. Therefore, for fixed values of $\{w_j\}_{j \neq i}$, the maximizer of (5) with respect to $w_i$ must lie within the interval $(-b_2, a_2]$.*

### 3.3 Optimal spatial resource allocation

In the previous subsection, both the locations and masses of the discrete measure were fixed, and the objective was to determine a service area partition. We now advance the analysis by formulating a quantization problem in which the locations and masses are jointly optimized to minimize the overall transport cost. This extension naturally connects to practical applications such as facility location and resource allocation. Specifically, we adopt a sequential optimization strategy: we first optimize the mass vector while holding the locations fixed, leveraging the optimality conditions established in Theorem 3.4, and subsequently update the locations to further reduce the aggregate loss.

**Theorem 3.5** (**Optimal masses given locations**). *Given fixed cardinality $M$ and fixed support points $y_1, \cdots, y_M \in X_2$, define*

$$Q(y_1, \cdots, y_M) := \min \left\{ \mathcal{E}^{\mathbf{k}_1, \mathbf{k}_2, k}(\mu_1, \mu_2) : \mu_2 = \sum_{i=1}^M m_i \delta_{y_i}, m_1, \cdots, m_M > 0 \right\}, \tag{11}$$

*where $\mathcal{E}^{\mathbf{k}_1, \mathbf{k}_2, k}(\mu_1, \mu_2)$ is defined in (2) with $\gamma$ runs over $\mathcal{M}(X_1 \times \{y_1, \ldots, y_M\})$. Assume that $\mu_1(C_i(\mathbf{0})) > 0$ for every $i = 1, \ldots, M$, where $\mathbf{0}$ is the $M$-dimensional zero vector. Then the minimizer of (11) is $(m_1^*, \cdots, m_M^*)$ satisfying $m_i^* = \mu_1(C_i(\mathbf{0})) > 0$ for every $i = 1, \ldots, M$. Moreover, we have*

$$Q(y_1, \cdots, y_M) = \sum_{i=1}^M \int_{C_i(\mathbf{0})} I_1(-b_1 \vee k \cdot c(x, y_i)) d\mu_1(x) + a_1 \mu_1(R). \tag{12}$$

To complete the quantization procedure, it remains to optimize over the locations of support points. Based on Theorem 3.5, this amounts to solving the following location optimization problem:

$$\min \{Q(y_1, \cdots, y_M) : y_1, \cdots, y_M \in X_2\},$$

where $Q(y_1, \cdots, y_M)$ is defined in (11). This optimization can be carried out efficiently using gradient-based methods [39, 40, 41], as the necessary gradients are derived and provided below.

**Proposition 3.2** (**Gradient of quantization objective function**). *Assume the cost function $c : X_1 \times X_2 \to (-\infty, +\infty]$ is differentiable with respect to its second argument, and denote its gradient by $\nabla_y c(x, y)$. Assume that $\mu_1(C_i(\mathbf{0})) > 0$ for every $i = 1, \ldots, M$. Then, for each $i = 1, \cdots, M$, the gradient of the quantization objective $Q(y_1, \cdots, y_M)$ with respect to $y_i$ is given by*

$$\nabla_{y_i} Q(y_1, \cdots, y_M) = k \int_{C_i(\mathbf{0}) \cap \{\mathbf{x}: \mathbf{k} \cdot \mathbf{c}(\mathbf{x}, \mathbf{y}_i) \in (-\mathbf{b}_1, \mathbf{a}_1)\}} \nabla_{y_i} c(x, y_i) d\mu_1(x).$$

# 4 Numerical examples and illustrative applications

Given our primary focus on establishing a rigorous theoretical framework and the space limitations, we illustrate the practical relevance of our results through two representative application scenarios with numerical examples. Further exploration of additional applications is left for future research.

## 4.1 Supply area division

We demonstrate the application of Section 3.2 through a spatial supply-area division problem. An urban planner must allocate $m = 4$ fixed emergency depots $Y = \{y_i\}_{i=1}^m \subset [0,1]^2$ to serve incident locations distributed continuously over $X \subset [0,1]^2$ with density $\mu$. The total incident mass is normalized to $M_\mu = 1$, while the available standby capacity is limited to $M_\nu = 0.7$, represented by the discrete measure $\nu = \sum_{i=1}^m \nu_i \delta_{y_i}$, with $\sum_i \nu_i = 0.7$. Travel cost is set to the Euclidean distance.

**Semi-discrete.** The demand distribution $\mu$ is continuous, while supply is concentrated at finitely many depots ($\nu$). The tessellation-based dual formulation thus applies directly.

**Unbalanced.** Since $M_\mu \neq M_\nu$, only a fraction of incidents can be served. The optimization identifies where to allocate supply and where to forgo service. We set $\mu$ to be uniform and initialize $\nu_i$ randomly.

**TV penalty.** Asymmetric newsvendor parameters are set to $a_1 = 1$, $b_1 = 0.5$, and $a_2 = b_2 = 0$, penalizing undersupply at rate $b_1$ and oversupply at rate $a_1$. We further let $k = 1$.

Proposition 3.1 provides the sub-gradient $g_i(w) = \text{clip}_{[-b_1, a_1]}(\nu_i - \mu(C_i(w)))$, which we use in gradient descent algorithms with decaying step size $\alpha_t = \alpha_0 (1+t)^{-\text{decay}}$, where $\alpha_0 = 0.05$ and $\text{decay} = 0.6$. The algorithm stops after 1000 iterations in 0.9 seconds. The final supply-area partition, determined by the optimized weights $\{w_i\}_{i=1}^4$, is shown in Figure 1. This example highlights the practicality of the dual formulation and subgradient structure from Section 3.2 in addressing semi-discrete, unbalanced, and asymmetrically penalized transport problems.

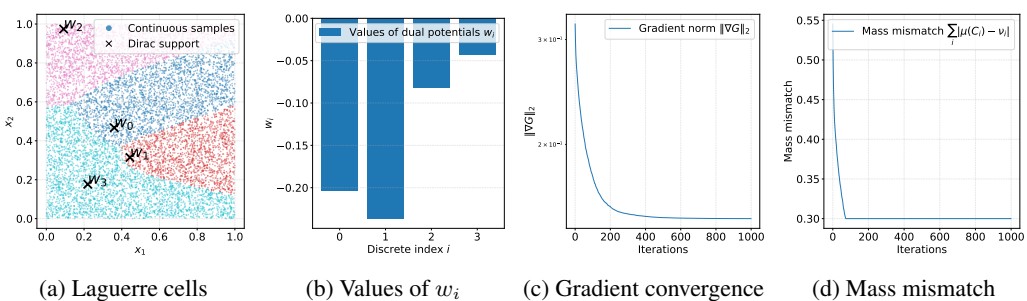

| (a) Laguerre cells | (b) Values of $w_i$ | (c) Gradient convergence | (d) Mass mismatch |

Figure 1: Fig. (a) displays the resulting supply-area partition, with black crosses indicating depot locations. Fig. (b) presents the learned dual weights $w_i \in [-1, 1]$. Figs. (c)–(d) plot the $\ell_2$ norm of the gradient and the total mass mismatch $\sum_i |\mu(C_i) - \nu_i|$.

## 4.2 Spatial resource allocation

This experiment extends the previous example by endogenously selecting the depots, echoing Section 3.3. Specifically, we now optimize the positions $Y = \{y_i\}_{i=1}^m \subset [0,1]^2$ along with the dual weights $w_i$, and the optimal mass allocation is determined by Theorem 3.5. Each point $x$ is assigned to the depot minimizing the effective cost $z(x, i) = c(x, y_i) - w_i$. If the minimal cost exceeds the threshold $a_1$, this point is assigned to the residual set $R$, incurring a penalty of $a_1 \mu(R)$. If a cell $C_i(w) = \{x : z(x, i) \leq z(x, j), \; \forall j\}$ becomes empty, the corresponding weight is set to $w_i = -b_2$, effectively deactivating the facility. The resulting outer objective, $Q(y) = \sum_i \int_{C_i(w^\star)} z(x, i) \, d\mu(x) + a_1 \mu(R)$, matches the form addressed in Theorem 3.5.

*Setting.* We sample $N = 2000$ incident locations from a two-component Gaussian mixture with means $(0.3, 0.3)$ and $(0.7, 0.7)$, and covariance $0.02 I_2$. Samples are clipped to $[0,1]^2$ and rescaled to ensure $M_\mu = 1$. The number of depots is fixed at $m = 4$, with initial positions drawn i.i.d. from the uniform distribution on $[0,1]^2$. Newsvendor parameters are set to $a_1 = 0.3$, $a_2 = 1$, $b_1 = 0.1$, and $b_2 = 0.2$. The choice of a smaller $a_1$ encourages visible residual mass, while the ordering

$b_1 < b_2 < a_1$ conforms to the structure of the folded loss $I_1$. A decaying step size $0.3/(1+t)^{0.6}$ is used. The CPU running time is 2.6 s, and the results shown in Figure 2 confirm that the outer–inner decomposition prescribed by Proposition 3.2 yields an effective gradient-based solver.

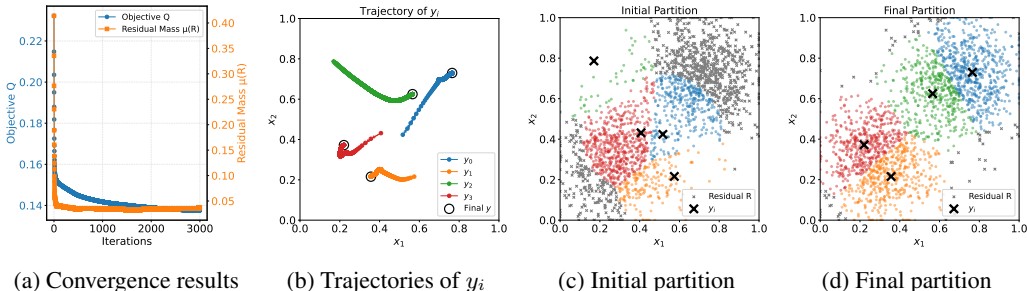

(a) Convergence results     (b) Trajectories of $y_i$     (c) Initial partition     (d) Final partition

Figure 2: Fig. (a) plots the objective $Q(y)$ (blue) and the residual mass $\mu(R)$ (orange), which stabilizes around 0.05, indicating that approximately $5\%$ of demand is optimally left unserved due to excessive transport cost. Fig. (b) illustrates the trajectories of the four depots: arrows trace movement from random initial positions toward demand clusters. In Fig. (c), the initial location of depots (black "×") and the nearby demands partition the entire area into several distinct parts (in different colors), which resembles a Laguerre diagram. In this partition, the demands within each Laguerre cell are serviced by the depot of the same color. However, there are still areas with high-density demands (gray "×") falling into the residual set $R$. After optimization, Fig. (d) shows four depots concentrating near high-density regions, while low-demand areas are absorbed into $R$, reflecting TV-induced saturation.

# 5 Conclusion and future work

This paper presents a comprehensive theoretical framework for unbalanced weak OT problems with total-variation penalties. We establish a general dual formulation that encompasses many existing models as special cases. We then focus on the semi-discrete setting, motivated by spatial resource allocation applications, and show that the problem admits a Laguerre tessellation structure. This allows for explicit optimality conditions and a next-level tractable quantization procedure. While our primary emphasis is on theoretical development, we include preliminary numerical examples to demonstrate the framework's applicability. We anticipate that this foundation will support future extensions in both theory and practice.

## Acknowledgments and Disclosure of Funding

Nhan-Phu Chung is funded by University of Economics Ho Chi Minh City, Vietnam. Jinhui Han is supported in part by the National Natural Science Foundation of China (Project "Data-Driven Operations Analytics Methods") and "The Fundamental Research Funds for the Central Universities, Peking University". The authors thank the anonymous reviewers for their comments.

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

## A   Proofs of assertions in Section 3

### A.1   Auxiliary lemmas

**Lemma A.1.** *Let $X$ be a metric space and $\mu_1, \mu_2, \nu_1, \nu_2 \in \mathcal{M}(X)$. Then*

$$|(\mu_1 - \nu_1)^+| + |(\mu_2 - \nu_2)^+| \geq |(\mu_1 + \mu_2 - \nu_1 - \nu_2)^+|.$$

*Proof.* The results follow from that

$$
\begin{aligned}
(\mu_1 - \nu_1)^+ + (\mu_2 - \nu_2)^+ &= \frac{1}{2}\Big[|\mu_1 - \nu_1| + |\mu_2 - \nu_2| + \mu_1 - \nu_1 + \mu_2 - \nu_2\Big] \\
&\geq \frac{1}{2}\Big[|\mu_1 + \mu_2 - (\nu_1 + \nu_2)| + \mu_1 + \mu_2 - (\nu_1 + \nu_2)\Big] \\
&= (\mu_1 + \mu_2 - (\nu_1 + \nu_2))^+.
\end{aligned}
$$

$\square$

**Lemma A.2.** *Let $X$ be a metric space and $\mu, \nu \in \mathcal{M}(X)$. Then*

$$|(\mu - \nu)^+| \geq (\mu(X) - \nu(X))^+.$$

*Proof.* Let $\nu = f\mu + \nu^\perp$ be the Lebesgue decomposition of $\nu$ with respect to $\mu$. Then

$$
\begin{aligned}
|(\mu - \nu)^+| &= |((1 - f)\mu - \nu^\perp)^+| \\
&= \int_X (1 - f)^+ d\mu \\
&= \frac{1}{2} \int_X (|1 - f| + 1 - f) d\mu \\
&\geq \frac{1}{2} \left( \left| \int_X (1 - f) d\mu \right| + \int_X (1 - f) d\mu \right) \\
&= \left( \int_X (1 - f) d\mu \right)^+ \\
&= (\mu(X) - \nu(X))^+.
\end{aligned}
$$

$\square$

**Lemma A.3.** *Let $c_1$ and $c_2$ be non-negative numbers. We define the map $J : \mathbb{R} \to (-\infty, +\infty]$ by*

$$
J(\phi) = \sup_{s > 0} \frac{\phi - c_1(1 - s)^+ - c_2(s - 1)^+}{s} = \begin{cases} +\infty & \text{if } \phi > c_1, \\ \phi & \text{if } -c_2 \leq \phi \leq c_1, \\ -c_2 & \text{otherwise.} \end{cases}
$$

*Then for every $\nu, \mu \in \mathcal{M}(X)$ we have that*

$$c_1|(\mu - \nu)^+| + c_2|(\nu - \mu)^+| \geq \int_X \varphi d\mu - \int_X J(\varphi) d\nu,$$

*for every $\varphi \in B(X)$ satisfying that $\sup_{x \in X} \varphi(x) \leq c_1$.*

*Proof.* The statement is true for the case $\mu = \nu$ are the null measure. Therefore, we can assume that $(\mu + \nu)(X) > 0$. Let $h$ be the Lebesgue density of $\nu$ with respect to $\mu + \nu$. We define $V := \{x \in X : 0 < h(x) < 1\}$, $V_\mu := \{x \in X : h(x) = 0\}$ and $V_\nu := \{x \in X : h(x) = 1\}$. Then

$(V, V_\mu, V_\nu)$ is a Borel partition of $X$. We define the Borel functions $f$ on $X$ by $f := \dfrac{h}{1-h}$ on $V$, and $f = 0$ on $X \setminus V$. Then $\nu = f\mu + \nu^\perp$, $\nu^\perp(X \setminus V_\nu) = \mu(V_\nu) = 0$. Then

$$c_1|(\mu - \nu)^+| + c_2|(\nu - \mu)^+| = c_1|[(1-f)\mu - \nu^\perp]^+ + c_2|[(f-1)\mu + \nu^\perp]^+|$$

$$= c_1 \int_{X \setminus V_\nu} (1-f)^+ d\mu + c_2 \left[ \int_{X \setminus V_\mu} (f-1)^+ d\mu + \nu^\perp(V_\nu) \right].$$

By the definition of $J$ we get that $c_1(1-f)^+ + c_2(f-1)^+ + fJ(\varphi) - \varphi \geq 0$. Hence for every $\varphi \in B(X)$ with $\sup_{x \in X} \varphi(x) \leq c_1$ we have that

$$c_1|(\mu-\nu)^+| + c_2|(\nu-\mu)^+| - \int_X \varphi d\mu + \int_X J(\varphi) d\nu$$

$$\geq c_1 \left[ \int_V (1-f)^+ d\mu + \int_{V_\mu} d\mu \right] + c_2 \left[ \int_V (f-1)^+ d\mu + \int_{V_\nu} d\nu^\perp \right] - \left( \int_V \varphi d\mu + \int_{V_\mu} \varphi d\mu \right)$$

$$+ \left( \int_V fJ(\varphi) d\mu + \int_{V_\nu} J(\varphi) d\nu^\perp \right)$$

$$= \int_V (c_1(1-f)^+ + c_2(f-1)^+ + fJ(\varphi) - \varphi) d\mu + \int_{V_\mu} (c_1 - \varphi) d\mu + \int_{V_\nu} (c_2 + J(\varphi)) d\nu^\perp$$

$$\geq 0.$$

$\square$

## A.2 Proof of Theorem 3.1

Before proving Theorem 3.1, let us provide some examples of the function $C : X_1 \times \mathcal{P}(X_2) \to (-\infty, +\infty]$ having property (T) (see Definition 3.1). Recall that for a metric space $X$, $\{\nu^n\}_{n \in \mathbb{N}} \subset \mathcal{M}(X)$ is tight if for every $\varepsilon > 0$, there exists a compact subset $K_\varepsilon$ of $X$ such that $\nu^n(X \setminus K_\varepsilon) < \varepsilon$ for every $n \in \mathbb{N}$.

**Example A.1.** *Let $c : X_1 \times X_2 \to (-\infty, +\infty]$ be a measurable function. Assume that $c$ has compact sublevels, i.e. for every $L \in \mathbb{R}$, the subset $\{(x_1, x_2) \in X_1 \times X_2 : c(x_1, x_2) \leq L\}$ of $X_1 \times X_2$ is compact. We define the map*

$$C : X_1 \times \mathcal{P}(X_2) \to (-\infty, +\infty]$$

*by $C(x_1, p) = \int_{X_2} c(x_1, x_2) dp(x_2)$ for every $x_1 \in X_1$, and $p \in \mathcal{P}(X_2)$. Given $M > 0$ and let $\{\gamma^n\}_{n \in \mathbb{N}} \subset \mathcal{M}(X_1 \times X_2)$ such that $\int_{X_1} C(x_1, \gamma^n_{x_1}) d\gamma^n_1(x_1) \leq M$, for every $n \in \mathbb{N}$. For every $L > 0$, the set*

$$A_L := \{(x_1, x_2) \in X_1 \times X_2 : c(x_1, x_2) \leq L\}$$

*is compact. Since*

$$\int_{X_1 \times X_2} c(x_1, x_2) d\gamma^n(x_1, x_2) = \int_{X_1} \int_{X_2} c(x_1, x_2) d\gamma^n_{x_1}(x_2) d\gamma^n_1(x_1) = \int_{X_1} C(x_1, \gamma^n_{x_1}) d\gamma^n_1(x_1),$$

*for every $L > 0$, we have*

$$\gamma^n((X_1 \times X_2) \setminus A_L) \leq \frac{\int_{X_1 \times X_2} c \, d\gamma^n}{L} = \frac{\int_{X_1} C(x_1, \gamma^n_{x_1}) d\gamma^n_1(x_1)}{L} \leq \frac{M}{L}.$$

*For every $\varepsilon > 0$, choose $L > 0$ such that $\dfrac{M}{L} < \varepsilon$, then $\gamma^n((X_1 \times X_2) \setminus A_L) < \varepsilon$, for every $n \in \mathbb{N}$. Therefore, the map $C$ has property (T).*

**Example A.2.** *If $C(x_1, p) = \alpha\left( \int_{X_2} c(x_1, x_2) dp(x_2) \right)$, for every $x_1 \in X_1$, $p \in \mathcal{P}(X_2)$ for some non-decreasing function $\alpha : (-\infty, +\infty] \to (-\infty, +\infty]$, and $c$ has compact sublevels, then $C$ has property (T).*

**Example A.3.** *If $X_1$ and $X_2$ are compact, then every measurable map $C : X_1 \times \mathcal{P}(X_2) \to (-\infty, +\infty]$ has property (T).*

*Proof of Theorem 3.1.* Let $\{\boldsymbol{\gamma}^n\}_{n \in \mathbb{N}} \subset \mathcal{M}(X_1 \times X_2)$ such that $\mathcal{E}_C(\mu_1, \mu_2) = \lim_{n \to \infty} \mathrm{ET}_C(\boldsymbol{\gamma}^n | \mu_1, \mu_2)$. First, we will prove that $\{\boldsymbol{\gamma}^n\}$ is bounded. Choose $t_0 \geq 0$ such that $\dfrac{F_1(s)}{s} \geq \dfrac{b_1}{2}$ and $\dfrac{F_2(s)}{s} \geq \dfrac{b_2}{2}$ for every $s \geq t_0$, where $F_i(s) = a_i(1-s)^+ + b_i(s-1)^+$. Now we will show that

$$\boldsymbol{\gamma}^n(X_1 \times X_2) \leq \frac{2}{M} \mathrm{ET}_C(\boldsymbol{\gamma}^n | \mu_1, \mu_2) \text{ for all } \boldsymbol{\gamma}^{n_j} \text{ with } \boldsymbol{\gamma}^n(X_1 \times X_2) \geq t_0 \max\{\mu_1(X_1), \mu_2(X_2)\},$$
(13)

where $M := \dfrac{b_1}{2} + \dfrac{b_2}{2} + \inf C > 0$. Let $\boldsymbol{\gamma}^n$ with $\boldsymbol{\gamma}^n(X_1 \times X_2) \geq t_0 \max\{\mu_1(X_1), \mu_2(X_2)\}$. If $\mu_1(X_1) = 0$ then $\mathcal{F}_1(\gamma_1^n | \mu_1) = b_1 \gamma_1^n(X_1)$. If $\mu_1(X_1) > 0$ then by the choice of $t_0$ and Lemma A.2 we get that

$$\begin{aligned}
\mathcal{F}_1(\gamma_1^n | \mu_1) &= a_1 |(\mu_1 - \gamma_1^n)^+| + b_1 |(\gamma_1^n - \mu_1)^+| \\
&\geq a_1 (\mu_1(X_1) - \gamma_1^n(X_1))^+ + b_1(\gamma_1^n(X_1) - \mu_1(X_1))^+ \\
&= \mu_1(X_1)\left[a_1\left(1 - \frac{\gamma_1^n(X_1)}{\mu_1(X_1)}\right)^+ + b_1\left(\frac{\gamma_1^n(X_1)}{\mu_1(X_1)} - 1\right)^+\right] \\
&= \gamma_1^n(X_1)\frac{\mu_1(X_1)}{\gamma_1^n(X_1)} F_1\left(\frac{\gamma_1^n(X_1)}{\mu_1(X_1)}\right) \\
&\geq \gamma_1^n(X_1)\frac{b_1}{2}.
\end{aligned}$$

Therefore, $\mathcal{F}_1(\gamma_1^n | \mu_1) \geq \boldsymbol{\gamma}^n(X_1 \times X_2)\dfrac{b_1}{2}$ for all $\boldsymbol{\gamma}^n$ with $\boldsymbol{\gamma}^n(X_1 \times X_2) \geq t_0 \max\{\mu_1(X_1), \mu_2(X_2)\}$. Similarly, $\mathcal{F}_2(\gamma_2^n | \mu_2) \geq \boldsymbol{\gamma}^n(X_1 \times X_2)\dfrac{b_2}{2}$ for all $\mu_2$ and $\boldsymbol{\gamma}^n$ with $\boldsymbol{\gamma}^n(X_1 \times X_2) \geq t_0 \max\{\mu_1(X_1), \mu_2(X_2)\}$. Then for all $\boldsymbol{\gamma}^n$ with $\boldsymbol{\gamma}^n(X_1 \times X_2) \geq t_0 \max\{\mu_1(X_1), \mu_2(X_2)\}$ we have that

$$\begin{aligned}
\mathrm{ET}_C^{\mathbf{k}_1, \mathbf{k}_2}(\boldsymbol{\gamma}^n | \mu_1, \mu_2) &= \mathcal{F}_1(\gamma_1^n | \mu_1) + \mathcal{F}_2(\gamma_2^n | \mu_2) + \int_{X_1} C(x_1, \gamma_{x_1}^n) d\gamma_1^n(x_1) \\
&\geq \boldsymbol{\gamma}^n(X_1 \times X_2)\left(\frac{b_1}{2} + \frac{b_2}{2} + \inf C\right).
\end{aligned}$$

Hence we get that $\{\boldsymbol{\gamma}^n\}_{n \in \mathbb{N}}$ is bounded.

$$\boldsymbol{\gamma}^n(X_1 \times X_2) \leq \frac{2}{M} \mathrm{ET}_C(\boldsymbol{\gamma}^n | \mu_1, \mu_2) \text{ for all } \boldsymbol{\gamma}^{n_j} \text{ with } \boldsymbol{\gamma}^n(X_1 \times X_2) \geq t_0 \max\{\mu_1(X_1), \mu_2(X_2)\},$$
(14)

Now we will prove that $\{\gamma_n\}_{n \in \mathbb{N}}$ is tight. There exists $K > 0$ such that $\mathrm{ET}_C^{\mathbf{a}, \mathbf{b}}(\boldsymbol{\gamma}^n | \mu_1, \mu_2) \leq K$ and hence $\int_{X_1} C(x_1, \gamma_{x_1}^n) d\gamma_1^n(x_1) \leq K$ for every $n$. Since $C$ has property (T), we get that $\{\boldsymbol{\gamma}^n\}$ is tight. As $\{\boldsymbol{\gamma}^n\}$ is tight and bounded, there exists $\boldsymbol{\gamma}^0 \in \mathcal{M}(X_1 \times X_2)$ and a subsequence of $\{\boldsymbol{\gamma}^n\}$ which is still denoted by $\{\boldsymbol{\gamma}^n\}$ such that $\boldsymbol{\gamma}^n$ converges weakly to $\boldsymbol{\gamma}^0$. From [42, Corollary 2.9] and [22, Lemma 4] we get that the map $\boldsymbol{\gamma} \mapsto \mathcal{E}_C(\boldsymbol{\gamma} | \mu_1, \mu_2)$ is lower semicontinuous. Therefore, $\boldsymbol{\gamma}^0$ is an optimal plan of problem (1). $\qquad\square$

**Remark A.1.** *If $C$ is nonnegative, we can prove the boundedness of $\{\boldsymbol{\gamma}^n\}$ easier as follows. Choose $M > 0$ such that $\mathrm{ET}_C^{\mathbf{a}, \mathbf{b}}(\boldsymbol{\gamma}^n | \mu_1, \mu_2) \leq M$ for every $n \in \mathbb{N}$. First, we will prove that $\{\boldsymbol{\gamma}^n\}$ is bounded. As $C$ is nonnegative we get that $a_1 |(\mu_1 - \gamma_1^n)^+| + b_1 |(\gamma_1^n - \mu_1)^+| \leq M$ for every $n \in \mathbb{N}$. Then $|\gamma_1^n - \mu_1|(X_1) = |(\mu_1 - \gamma_1^n)^+| + |(\gamma_1^n - \mu_1)^+|$ is bounded. Hence $\gamma_1^n(X_1)$ is bounded. Therefore $\{\boldsymbol{\gamma}^n\}$ is bounded.*

### A.3 Proof of Theorem 3.2

**Lemma A.4.** *We have*

$$\sup_{(\varphi_1, \varphi_2) \in \Phi_J} \sum_{i=1}^{2} \int_{X_i} \varphi_i(x_i) d\mu_i(x_i) = \sup_{(\varphi_1, \varphi_2) \in \Phi_I} \sum_{i=1}^{2} \int_{X_i} I_i(\varphi_i(x_i)) d\mu_i(x_i).$$
(15)

*Proof.* For any $(\varphi_1, \varphi_2) \in \Phi_J$, we have $-b_1 \leq J_1(\varphi_1(x_1)) \leq a_1, -b_2 \leq J_2(\varphi_2(x_2)) \leq a_2$ and $J_1(\varphi_1(x_1)) + p(J_2(\varphi_2)) \leq C(x_1, p)$ for every $x_1 \in X_1, p \in \mathcal{P}(X_2)$. Since $J_1$ and $J_2$ are continuous on $(-\infty, a_1]$ and $(-\infty, a_2]$, respectively, one has $(J_1(\varphi_1), J_2(\varphi_2)) \in \Phi_I$. Therefore,

$$\sum_{i=1}^{2} \int_{X_i} \varphi_i(x_i) d\mu_i(x_i) \leq \sum_{i=1}^{2} \int_{X_i} J_i(\varphi_i(x_i)) d\mu_i(x_i)$$

$$= \sum_{i=1}^{2} \int_{X_i} I_i \circ J_i(\varphi_i(x_i)) d\mu_i(x_i)$$

$$\leq \sup_{(\varphi_1, \varphi_2) \in \Phi_I} \sum_{i=1}^{2} \int_{X_i} I_i(\varphi_i(x_i)) d\mu_i(x_i).$$

On the other hand, for any $(\varphi_1, \varphi_2) \in \Phi_I$, we have $-b_1 \leq I_1(\varphi_1(x_1)) \leq a_1, -b_2 \leq I_2(\varphi_2(x_2)) \leq a_2$ and hence $J_1 \circ I_1(\varphi_1(x_1)) + p(J_2 \circ I_2(\varphi_2)) \leq C(x_1, p)$ for every $x_1 \in X_1, p \in \mathcal{P}(X_2)$. Since $I_1$ and $I_2$ are continuous on $[-b_1, \infty)$ and $[-b_2, \infty)$, respectively, one has $(I_1(\varphi_1), I_2(\varphi_2)) \in \Phi_J$. Hence,

$$\sum_{i=1}^{2} \int_{X_i} I_i(\varphi_i(x_i)) d\mu_i(x_i) \leq \sup_{(\varphi_1, \varphi_2) \in \Phi_J} \sum_{i=1}^{2} \int_{X_i} \varphi_i(x_i) d\mu_i(x_i).$$

We obtain the result. $\qquad\square$

Given a metric space $X$, for every $\mu \in \mathcal{M}(X)$, the map $T_\mu : C_b(X) \to \mathbb{R}$, defined by $f \mapsto \int_X f d\mu$, is a bounded linear operator, i.e., it belongs to $(C_b(X))^*$. We define the functional ET $: (C_b(X_1))^* \times (C_b(X_2))^* \to [-\infty, +\infty]$ as follows.

$$\mathrm{ET}^{\mathbf{k}_1, \mathbf{k}_2}(T_1, T_2) := \begin{cases} \mathcal{E}_C(\mu_1, \mu_2) & \text{if } (T_1, T_2) = (T_{\mu_1}, T_{\mu_2}), \\ +\infty & \text{otherwise,} \end{cases}$$

Given $\mu, \nu \in \mathcal{M}(X)$, if $\int_X f d\mu = \int_X f d\nu$ for every $f \in C_b(X)$ then one gets $\mu = \nu$ [48, Theorem 5.9, page 39]. Therefore, for every metric space $X$ we can consider $\mathcal{M}(X)$ as a subset of $(C_b(X))^*$. Hence, the map $\mathrm{ET}^{\mathbf{k}_1, \mathbf{k}_2, k}$ is well defined.

For convenience, we will write $\mathrm{ET}^{\mathbf{k}_1, \mathbf{k}_2}(\mu_1, \mu_2)$ for $\mathrm{ET}^{\mathbf{k}_1, \mathbf{k}_2}(T_{\mu_1}, T_{\mu_2})$ for every $(\mu_1, \mu_2) \in \mathcal{M}(X_1) \times \mathcal{M}(X_2)$. The weak topology on $\mathcal{M}(X)$ is the smallest topology such that for each $f \in C_b(X)$, the map $\mu \mapsto \int_X f d\mu$ is continuous, i.e. a sequence $\{\mu_n\}_{n \in \mathbb{N}} \subset \mathcal{M}(X)$ converges weakly to $\mu \in \mathcal{M}(X)$ if and only if $\lim_{n \to \infty} \int_X f d\mu_n = \int_X f d\mu$ for every $f \in C_b(X)$.

**Lemma A.5.** *Let $X_1, X_2$ be Polish metric spaces. Then*

1. *the functional $\mathrm{ET}^{\mathbf{k}_1, \mathbf{k}_2} : (C_b(X_1))^* \times (C_b(X_2))^* \to (-\infty, +\infty]$ is convex and positively one homogeneous, i.e. $\mathrm{ET}^{\mathbf{k}_1, \mathbf{k}_2}(\lambda T_1, \lambda T_2) = \lambda \mathrm{ET}^{\mathbf{a}, \mathbf{b}}(T_1, T_2)$ for every $\lambda \geq 0, T_1 \in (C_b(X_1))^*, T_2 \in (C_b(X_2))^*$;*

2. *Assume that $c$ has compact sublevels in $X_1 \times X_2$. Then the function $\mathrm{ET}^{\mathbf{k}_1, \mathbf{k}_2}$ is lower semicontinuous under the weak topology.*

*Proof.* **Claim 1:** Using the convention that $0 \cdot (+\infty) = 0$, it is clear that $\mathrm{ET}^{\mathbf{k}_1, \mathbf{k}_2}(0, 0) = 0$ and $\mathrm{ET}^{\mathbf{a}, \mathbf{b}}(\lambda T_1, \lambda T_2) = \lambda \mathrm{ET}^{\mathbf{k}_1, \mathbf{k}_2}(T_1, T_2)$ for every $\lambda \geq 0, (T_1, T_2) \notin \mathcal{M}(X_1) \times \mathcal{M}(X_2)$. Hence, to check $\mathrm{ET}^{\mathbf{k}_1, \mathbf{k}_2}$ is positively one homogeneous it is sufficient to check $\mathrm{ET}^{\mathbf{k}_1, \mathbf{k}_2}(\lambda T_{\mu_1}, \lambda T_{\mu_2}) = \lambda \mathrm{ET}^{\mathbf{k}_1, \mathbf{k}_2}(T_{\mu_1}, T_{\mu_2})$ for every $\lambda > 0, (\mu_1, \mu_2) \in \mathcal{M}(X_1) \times \mathcal{M}(X_2)$. Given $\gamma \in \mathcal{M}(X_1 \times X_2)$ and

$\lambda > 0$. We have that

$$
\begin{aligned}
\mathrm{ET}^{\mathbf{k}_1,\mathbf{k}_2}(\lambda T_{\mu_1}, \lambda T_{\mu_2}) =& \mathrm{ET}^{\mathbf{k}_1,\mathbf{k}_2}(T_{\lambda\mu_1}, T_{\lambda\mu_2}) = \mathcal{E}_C^{\mathbf{k}_1,\mathbf{k}_2}(\lambda\mu_1, \lambda\mu_2) \\
=& \inf\{\mathrm{ET}^{\mathbf{k}_1,\mathbf{k}_2}(\boldsymbol{\gamma}|\lambda\mu_1, \lambda\mu_2) : \boldsymbol{\gamma} \in \mathcal{M}(X_1 \times X_2)\} \\
=& \inf\{\mathrm{ET}^{\mathbf{k}_1,\mathbf{k}_2}(\lambda\boldsymbol{\gamma}|\lambda\mu_1, \lambda\mu_2) : \boldsymbol{\gamma} \in \mathcal{M}(X_1 \times X_2)\} \\
=& \inf\Big\{a_1|(\lambda\mu_1 - \lambda\gamma_1)^+| + b_1|(\lambda\gamma_1 - \lambda\mu_1)^+| + a_2|(\lambda\mu_2 - \lambda\gamma_2)^+| + \\
& + b_2|(\lambda\gamma_2 - \lambda\mu_2)^+| + \lambda\int_{X_1} C(x_1, \gamma_{x_1})d\gamma_1(x_1) : \boldsymbol{\gamma} \in \mathcal{M}(X_1 \times X_2)\Big\} \\
=& \lambda\inf\Big\{a_1|(\mu_1 - \gamma_1)^+| + b_1|(\gamma_1 - \mu_1)^+| + a_2|(\mu_2 - \gamma_2)^+| + \\
& + b_2|(\gamma_2 - \mu_2)^+| + \int_{X_1} C(x_1, \gamma_{x_1})d\gamma_1(x_1) : \boldsymbol{\gamma} \in \mathcal{M}(X_1 \times X_2)\Big\} \\
=& \lambda\mathrm{ET}^{\mathbf{k}_1,\mathbf{k}_2}(\mu_1, \mu_2) \\
=& \lambda\mathrm{ET}^{\mathbf{k}_1,\mathbf{k}_2}(T_{\mu_1}, T_{\mu_2}).
\end{aligned}
$$

By the homogeneity property of $\mathrm{ET}^{\mathbf{k}_1,\mathbf{k}_2}$, to show that $\mathrm{ET}^{\mathbf{k}_1,\mathbf{k}_2}$ is convex, we onlyneed to check that

$$\mathrm{ET}^{\mathbf{k}_1,\mathbf{k}_2}(\mu_1, \mu_2) + \mathrm{ET}(\nu_1, \nu_2) \geq \mathrm{ET}^{\mathbf{k}_1,\mathbf{k}_2}(\mu_1 + \nu_1, \mu_2 + \nu_2) \text{ for every } \mu_i, \nu_i \in \mathcal{M}(X_i), i = 1, 2.$$

We will consider $(\mu_1, \mu_2), (\nu_1, \nu_2) \in \mathcal{M}(X_1) \times \mathcal{M}(X_2)$ such that $\mathrm{ET}^{\mathbf{k}_1,\mathbf{k}_2}(\mu_1, \mu_2) < \infty$ and $\mathrm{ET}^{\mathbf{k}_1,\mathbf{k}_2}(\nu_1, \nu_2) < \infty$ (the other cases are trivial). Let $\{\boldsymbol{\gamma}^n\}_{n\in\mathbb{N}}, \{\overline{\boldsymbol{\gamma}}^n\}_{n\in\mathbb{N}} \subset \mathcal{M}(X_1 \times X_2)$ such that $\mathrm{ET}^{\mathbf{k}_1,\mathbf{k}_2}(\mu_1, \mu_2) = \lim_{n\to\infty}\mathrm{ET}^{\mathbf{k}_1,\mathbf{k}_2}(\boldsymbol{\gamma}^n|\mu_1, \mu_2)$ and $\mathrm{ET}^{\mathbf{k}_1,\mathbf{k}_2}(\nu_1, \nu_2) = \lim_{n\to\infty}\mathrm{ET}^{\mathbf{k}_1,\mathbf{k}_2}(\overline{\boldsymbol{\gamma}}^n|\nu_1, \nu_2)$.

As $\left((d\gamma_1^n/d(\gamma_1^n + \overline{\gamma}_1^n))\gamma_{x_1}^n + (d\overline{\gamma}_1^n/d(\gamma_1^n + \overline{\gamma}_1^n))\overline{\gamma}_{x_1}^n\right)_{x_1 \in X_1}$ is the disintegration of $\boldsymbol{\gamma}^n + \overline{\boldsymbol{\gamma}}^n$ with respect to $\gamma_1^n + \overline{\gamma}_1^n$ and $C(x_1, \cdot)$ is convex on $\mathcal{P}(X_2)$ for every $x_1 \in X_1$, we obtain that

$$\int_{X_1} C(x_1, \gamma_{x_1}^n)d\gamma_1^n(x_1) + \int_{X_1} C(x_1, \overline{\gamma}_{x_1}^n)d\overline{\gamma}_1^n(x_1) \geq \int_{X_1} C(x_1, (\boldsymbol{\gamma}^n + \overline{\boldsymbol{\gamma}}^n)_{x_1})d(\gamma_1^n + \overline{\gamma}_1^n)(x_1).$$

Combining with Lemma A.1, we have that

$$
\begin{aligned}
& \mathrm{ET}^{\mathbf{k}_1,\mathbf{k}_2}(\mu_1, \mu_2) + \mathrm{ET}^{\mathbf{k}_1,\mathbf{k}_2}(\nu_1, \nu_2) \\
=& \lim_{n\to\infty}\Big[a_1|(\mu_1 - \gamma_1^n)^+| + b_1|(\gamma_1^n - \mu_1)^+| + a_2|(\mu_2 - \gamma_2^n)^+| + b_2|(\gamma_2^n - \mu_2)^+| + \int_{X_1} C(x_1, \gamma_{x_1}^n)d\gamma_1^n(x_1) + \\
& + a_1|(\nu_1 - \overline{\gamma}_1^n)^+| + b_1|(\overline{\gamma}_1^n - \nu_1)^+| + a_2|(\nu_2 - \overline{\gamma}_2^n)^+| + b_2|(\overline{\gamma}_2^n - \nu_2)^+| + \int_{X_1} C(x_1, \overline{\gamma}_{x_1}^n)d\overline{\gamma}_1^n(x_1)\Big] \\
\geq& \lim_{n\to\infty}\Big[a_1|(\mu_1 + \nu_1 - (\gamma_1^n + \overline{\gamma}_1^n))^+| + b_1|(\gamma_1^n + \overline{\gamma}_1^n - (\mu_1 + \nu_1)^+| + a_2|(\mu_1 + \nu_1 - (\gamma_1^n + \overline{\gamma}_1^n))^+| \\
& + b_2|(\gamma_1^n + \overline{\gamma}_1^n - (\mu_1 + \nu_1)^+| + \int_{X_1} C(x_1, (\boldsymbol{\gamma}^n + \overline{\boldsymbol{\gamma}}^n)_{x_1})d(\gamma_1^n + \overline{\gamma}_1^n)(x_1)\Big] \\
\geq& \mathrm{ET}^{\mathbf{k}_1,\mathbf{k}_2}(\mu_1 + \nu_1, \mu_2 + \nu_2).
\end{aligned}
$$

Therefore, $\mathrm{ET}^{\mathbf{k}_1,\mathbf{k}_2}$ is convex.

**Claim 2:** For $i = 1, 2$, let $\{\mu_i^n\} \subset \mathcal{M}(X_i)$ such that $\mu_i^n$ weakly converges to $\mu_i$ as $n \to \infty$.

To prove $\mathrm{ET}^{\mathbf{k}_1,\mathbf{k}_2}$ is lower semicontinuous, we need to show that

$$\liminf_{n\to\infty} \mathrm{ET}^{\mathbf{k}_1,\mathbf{k}_2}(\mu_1^n, \mu_2^n) \geq \mathrm{ET}^{\mathbf{k}_1,\mathbf{k}_2}(\mu_1, \mu_2).$$

We only need to check it for the case $\liminf_{n\to\infty}\mathrm{ET}^{\mathbf{k}_1,\mathbf{k}_2}(\mu_1^n, \mu_2^n) < \infty$. We can choose a subsequence $\{n_j\}_{j\in\mathbb{N}}$ such that $\liminf_{n\to\infty}\mathrm{ET}^{\mathbf{k}_1,\mathbf{k}_2}(\mu_1^n, \mu_2^n) = \lim_{j\to\infty}\mathrm{ET}^{\mathbf{k}_1,\mathbf{k}_2}(\mu_1^{n_j}, \mu_2^{n_j})$. For every $j \in \mathbb{N}$, applying Theorem 3.1 there exists $\boldsymbol{\gamma}^{n_j} \in \mathcal{M}(X_1 \times X_2)$ such that

$$\mathrm{ET}^{\mathbf{k}_1,\mathbf{k}_2}(\mu_1^{n_j}, \mu_2^{n_j}) = \mathrm{ET}^{\mathbf{k}_1,\mathbf{k}_2}(\boldsymbol{\gamma}^{n_j}|\mu_1^{n_j}, \mu_2^{n_j}) \text{ for every } j \in \mathbb{N}.$$

Then there exists $K > 0$ such that $\text{ET}^{\mathbf{k}_1, \mathbf{k}_2}(\boldsymbol{\gamma}^{n_j} | \mu_1^{n_j}, \mu_2^{n_j}) \leq K$ for every $j$. Hence $\int_{X_1} C(x_1, \gamma_{x_1}^{n_j}) d\gamma_1^{n_j}(x_1) \leq K$ for every $j$. Since $C$ has property (T) we get that $\{\boldsymbol{\gamma}^{n_j}\}$ is tight.

Now we will prove that $\{\boldsymbol{\gamma}^{n_j}\}$ is bounded. Choose $t_0 \geq 0$ such that $\dfrac{F_1(s)}{s} \geq \dfrac{b_1}{2}$ and $\dfrac{F_2(s)}{s} \geq \dfrac{b_2}{2}$ for every $s \geq t_0$. Now we will show that

$$\boldsymbol{\gamma}^{n_j}(X_1 \times X_2) \leq \frac{2}{M} \text{ET}^{\mathbf{k}_1, \mathbf{k}_2}(\boldsymbol{\gamma}^{n_j} | \mu_1^{n_j}, \mu_2^{n_j}) \text{ for all } \boldsymbol{\gamma}^{n_j} \text{ with } \boldsymbol{\gamma}^{n_j}(X_1 \times X_2) \geq t_0 \max\{\mu_1^{n_j}(X_1), \mu_2^{n_j}(X_2)\},$$
(16)

where $M := \dfrac{b_1}{2} + \dfrac{b_2}{2} + \inf C > 0$. Let $\boldsymbol{\gamma}^{n_j}$ with $\boldsymbol{\gamma}^{n_j}(X_1 \times X_2) \geq t_0 \max\{\mu_1^{n_j}(X_1), \mu_2^{n_j}(X_2)\}$. If $\mu_1^{n_j}(X_1) = 0$ then $\mathcal{F}_1(\gamma_1^{n_j} | \mu_1^{n_j}) = b_1 \gamma_1^{n_j}(X_1)$. If $\mu_1^{n_j}(X_1) > 0$ then by the choice of $t_0$ and Lemma A.2 we get that

$$\begin{aligned}
\mathcal{F}_1(\gamma_1^{n_j} | \mu_1^{n_j}) &= a_1 |(\mu_1^{n_j} - \gamma_1^{n_j})^+| + b_1 |(\gamma_1^{n_j} - \mu_1^{n_j})^+| \\
&\geq a_1 (\mu_1^{n_j}(X_1) - \gamma_1^{n_j}(X_1))^+ + b_1 (\gamma_1^{n_j}(X_1) - \mu_1^{n_j}(X_1))^+ \\
&\geq \mu_1^{n_j}(X_1) \left[ a_1 \left( 1 - \frac{\gamma_1^{n_j}(X_1)}{\mu_1^{n_j}(X_1)} \right)^+ + b_1 \left( \frac{\gamma_1^{n_j}(X_1)}{\mu_1^{n_j}(X_1)} - 1 \right)^+ \right] \\
&= \gamma_1^{n_j}(X_1) \frac{\mu_1^{n_j}(X_1)}{\gamma_1^{n_j}(X_1)} F_1 \left( \frac{\gamma_1^{n_j}(X_1)}{\mu_1^{n_j}(X_1)} \right) \\
&\geq \gamma_1^{n_j}(X_1) \frac{b_1}{2}.
\end{aligned}$$

Therefore, $\mathcal{F}_1(\gamma_1^{n_j} | \mu_1^{n_j}) \geq \boldsymbol{\gamma}^{n_j}(X_1 \times X_2) \dfrac{b_1}{2}$ for all $\mu_1^{n_j}$ and $\boldsymbol{\gamma}^{n_j}$ with $\boldsymbol{\gamma}^{n_j}(X_1 \times X_2) \geq t_0 \max\{\mu_1^{n_j}(X_1), \mu_2^{n_j}(X_2)\}$. Similarly, $\mathcal{F}_2(\gamma_2^{n_j} | \mu_2^{n_j}) \geq \boldsymbol{\gamma}^{n_j}(X_1 \times X_2) \dfrac{b_2}{2}$ for all $\mu_2^{n_j}$ and $\boldsymbol{\gamma}^{n_j}$ with $\boldsymbol{\gamma}^{n_j}(X_1 \times X_2) \geq t_0 \max\{\mu_1^{n_j}(X_1), \mu_2^{n_j}(X_2)\}$. Then for all $\boldsymbol{\gamma}^{n_j}$ with $\boldsymbol{\gamma}^{n_j}(X_1 \times X_2) \geq t_0 \max\{\mu_1^{n_j}(X_1), \mu_2^{n_j}(X_2)\}$ we have that

$$\begin{aligned}
\text{ET}^{\mathbf{k}_1, \mathbf{k}_2}(\boldsymbol{\gamma}^{n_j} | \mu_1^{n_j}, \mu_2^{n_j}) &= \mathcal{F}_1(\gamma_1^{n_j} | \mu_1^{n_j}) + \mathcal{F}_2(\gamma_2^{n_j} | \mu_2^{n_j}) + \int_{X_1} C(x_1, \gamma_{x_1}^n) d\gamma_1^n(x_1) \\
&\geq \boldsymbol{\gamma}^{n_j}(X_1 \times X_2) \left( \frac{b_1}{2} + \frac{b_2}{2} + \inf C \right)
\end{aligned}$$

Hence, we get (16). On the other hand, $\{\mu_1^{n_j}(X_1)\}_{j \in \mathbb{N}}$ and $\{\mu_1^{n_j}(X_2)\}_{j \in \mathbb{N}}$ are bounded as $\mu_i^{n_j}$ weakly converges to $\mu_i$ for $i = 1, 2$. Therefore $\{\boldsymbol{\gamma}^{n_j}\}_{j \in \mathbb{N}}$ is bounded. As $\{\boldsymbol{\gamma}^{n_j}\}_{j \in \mathbb{N}}$ is also tight, applying Prokhorov's theorem there exists a subsequence $\{\boldsymbol{\gamma}^{n_j}\}_{j \in \mathbb{N}}$, still denoted by $\{\boldsymbol{\gamma}^{n_j}\}_{j \in \mathbb{N}}$, and a $\boldsymbol{\gamma} \in \mathcal{M}(X_1 \times X_2)$ such that $\boldsymbol{\gamma}^{n_j}$ weakly converges to $\boldsymbol{\gamma}$. Applying [13, Theorem 8.4.7] or [61, Part 2, Theorem 3] we get that

$$\liminf_{j \to \infty} |\mu_1^{n_j} - \gamma_1^{n_j}|(X_1) \geq |\mu_1 - \gamma_1|(X_1).$$

Hence

$$\begin{aligned}
\liminf_{j \to \infty} |(\mu_1^{n_j} - \gamma_1^{n_j})^+| &= \liminf_{j \to \infty} \frac{1}{2} \left[ |\mu_1^{n_j} - \gamma_1^{n_j}|(X_1) + (\mu_1^{n_j} - \gamma_1^{n_j})(X_1) \right] \\
&\geq \frac{1}{2} \left[ |\mu_1 - \gamma_1|(X_1) + (\mu_1 - \gamma_1)(X_1) \right] \\
&= |(\mu_1 - \gamma_1)^+|.
\end{aligned}$$

Similarly, we have

$$\begin{aligned}
\liminf_{j \to \infty} |(\mu_2^{n_j} - \gamma_2^{n_j})^+| &\geq |(\mu_2 - \gamma_2)^+|, \\
\liminf_{j \to \infty} |(\gamma_1^{n_j} - \mu_1^{n_j})^+| &\geq |(\gamma_1 - \mu_1)^+|, \\
\liminf_{j \to \infty} |(\gamma_2^{n_j} - \mu_2^{n_j})^+| &\geq |(\gamma_2 - \mu_2)^+|.
\end{aligned}$$

Then

$$
\liminf_{n \to \infty} \mathrm{ET}^{\mathbf{k}_1, \mathbf{k}_2}(\mu_1^n, \mu_2^n) = \liminf_{n \to \infty} \mathrm{ET}^{\mathbf{k}_1, \mathbf{k}_2}(\gamma^n | \mu_1^n, \mu_2^n)
$$

$$
= \liminf_{n \to \infty} \left[ a_1 |(\mu_1^{n_j} - \gamma_1^{n_j})^+| + b_1 |(\gamma_1^{n_j} - \mu_1^{n_j})^+| + a_2 |(\mu_2^{n_j} - \gamma_2^{n_j})^+| \right.
$$

$$
\left. + b_2 |(\gamma_2^{n_j} - \mu_2^{n_j})^+| + \int_{X_1} C(x_1, \gamma_{x_1}^{n_j}) d\gamma_1^{n_j}(x_1) \right]
$$

$$
\geq a_1 |(\mu_1 - \gamma_1)^+| + b_1 |(\gamma_1 - \mu_1)^+| + a_2 |(\mu_2 - \gamma_2)^+|
$$

$$
+ b_2 |(\gamma_2 - \mu_2)^+| + \int_{X_1} C(x_1, \gamma_{x_1}^{n_j}) d\gamma_1^{n_j}(x_1)
$$

$$
\geq \mathrm{ET}^{\mathbf{k}_1, \mathbf{k}_2}(\mu_1, \mu_2).
$$

$\square$

*Proof of Theorem 3.2.* We denote by $(\mathrm{ET}^{\mathbf{k}_1, \mathbf{k}_2})^*$ the Fenchel conjugate of $\mathrm{ET}^{\mathbf{k}_1, \mathbf{k}_2}$, i.e.

$$
(\mathrm{ET}^{\mathbf{k}_1, \mathbf{k}_2})^*(\varphi_1, \varphi_2) := \sup_{\nu_1 \in \mathcal{M}(X_1), \nu_2 \in \mathcal{M}(X_2)} \left\{ \sum_{i=1}^{2} \int \varphi_i d\nu_i(x_i) - \mathrm{ET}^{\mathbf{k}_1, \mathbf{k}_2}(\nu_1, \nu_2) \right\},
$$

for every $\varphi_i \in C_b(X_i)$, $i = 1, 2$. By Lemma A.5 we get that

$$
(\mathrm{ET}^{\mathbf{k}_1, \mathbf{k}_2})^*(\varphi_1, \varphi_2) = \begin{cases} 0 & \text{if } (\varphi_1, \varphi_2) \in \Phi_E, \\ +\infty & \text{otherwise,} \end{cases}
$$

where

$$
\Phi_E := \left\{ (\varphi_1, \varphi_2) \in C_b(X_1) \times C_b(X_2) : \sum_{i=1}^{2} \int_{X_i} \varphi_i(x_i) d\nu_i(x_i) \leq \mathrm{ET}^{\mathbf{k}_1, \mathbf{k}_2}(\nu_1, \nu_2) \right.
$$

$$
\left. \text{for every } (\nu_1, \nu_2) \in \mathcal{M}(X_1) \times \mathcal{M}(X_2) \right\}.
$$

We now check that $\Phi_J \subset \Phi_E$. Let any $(\varphi_1, \varphi_2) \in \Phi_J$. Let $\nu_i \in \mathcal{M}(X_i)$, $i = 1, 2$. If $\mathcal{E}_C^{\mathbf{k}_1, \mathbf{k}_2}(\nu_1, \nu_2) = +\infty$ then it is clear that $\sum_{i=1}^{2} \int_{X_i} \varphi_i(x_i) d\nu_i(x_i) \leq \mathrm{ET}^{\mathbf{k}_1, \mathbf{k}_2}(\nu_1, \nu_2)$. Thus, we only consider $\mathcal{E}_C^{\mathbf{k}_1, \mathbf{k}_2}(\nu_1, \nu_2) < +\infty$. Applying Theorem 3.1 there exists $\gamma \in \mathcal{M}(X_1 \times X_2)$ such that $\mathcal{E}_C^{\mathbf{k}_1, \mathbf{k}_2}(\nu_1, \nu_2) = \mathrm{ET}^{\mathbf{k}_1, \mathbf{k}_2}(\gamma | \nu_1, \nu_2)$. Then, applying Lemma A.3 we get that

$$
\mathrm{ET}^{\mathbf{k}_1, \mathbf{k}_2}(\nu_1, \nu_2)
$$

$$
= a_1 |(\nu_1 - \gamma_1)^+| + b_1 |(\gamma_1 - \nu_1)^+| + a_2 |(\nu_2 - \gamma_2)^+| + b_2 |(\gamma_2 - \mu_2)^+| + \int_{X_1} C(x_1, \gamma_{x_1}) d\gamma_1(x_1)
$$

$$
\geq \int_{X_1} \varphi_1 d\nu_1 - \int_{X_1} J_1(\varphi_1) d\gamma_1 + \int_{X_2} \varphi_2 d\nu_2 - \int_{X_2} J_2(\varphi_2) d\gamma_2 + \int_{X_1} [J_1(\varphi_1(x_1)) + \gamma_{x_1}(J_2(\varphi_2))] d\gamma_1(x_1)
$$

$$
= \int_{X_1} \varphi_1 d\nu_1 - \int_{X_1} J_1(\varphi_1) d\gamma_1 + \int_{X_2} \varphi_2 d\nu_2 - \int_{X_2} J_2(\varphi_2) d\gamma_2 + \int_{X_1} J_1(\varphi_1) d\gamma_1 +
$$

$$
+ \int_{X_1} \int_{X_2} J_2(\varphi_2(x_2)) d\gamma_{x_1}(x_2) d\gamma_1(x_1)
$$

$$
= \int_{X_1} \varphi_1 d\nu_1 + \int_{X_2} \varphi_2 d\nu_2 - \int_{X_2} J_2(\varphi_2) d\gamma_2 + \int_{X_2} J_2(\varphi_2(x_2)) d\gamma_2(x_2)
$$

$$
= \int_{X_1} \varphi_1 d\nu_1 + \int_{X_2} \varphi_2 d\nu_2.
$$

Therefore $(\varphi_1, \varphi_2) \subset \Phi_E$. Hence $\Phi_J \subset \Phi_E$.

Now, let $(\varphi_1, \varphi_2) \in \Phi_E$. We will show that $(\varphi_1, \varphi_2) \in \Phi_J$. Denote by $\eta$ the null measure on $X_1 \times X_2$. As $(\varphi_1, \varphi_2) \in \Phi_E$, for every $(\nu_1, \nu_2) \in \mathcal{M}(X_1) \times \mathcal{M}(X_2)$ one has

$$
\sum_{i=1}^{2} \int_{X_i} \varphi_i(x_i) d\nu_i(x_i) \leq \mathrm{ET}^{\mathbf{k}_1, \mathbf{k}_2}(\nu_1, \nu_2) \leq \mathcal{E}_C^{\mathbf{k}_1, \mathbf{k}_2}(\eta | \nu_1, \nu_2) = a_1 |\nu_1| + a_2 |\nu_2|.
$$

For every $z \in X_1$, setting $\nu_1 := \delta_z$ and $\nu_2$ is the null measure on $X_2$, we obtain that $\varphi_1(z) \leq a_1$. Similarly, we also have $\varphi_2 \leq a_2$ on $X_2$.

Let $x_1 \in X_1$ and $p \in \mathcal{P}(X_2)$. Now we will prove that $J_1(\varphi_1(x_1)) + p(J_2(\varphi_2)) \leq C(x_1, p)$. For every $r > 0$, put $\nu_1 := \delta_{x_1}$ and $\boldsymbol{\gamma} := r\delta_{x_1} \otimes p$. Then for every $\nu_2 \in \mathcal{M}(X_2)$, we have

$$
\begin{aligned}
\varphi_1(x_1) + \int_{X_2} \varphi_2 d\nu_2 &= \sum_{i=1}^{2} \int_{X_i} \varphi_i d\nu_i \\
&\leq \mathrm{ET}^{\mathbf{k}_1, \mathbf{k}_2}(\nu_1, \nu_2) \\
&\leq \mathcal{E}_C^{\mathbf{k}_1, \mathbf{k}_2}(\boldsymbol{\gamma}|\nu_1, \nu_2) \\
&= a_1(1-r)^+ + b_1(r-1)^+ + a_2|(\nu_2 - \gamma_2)^+| + b_2|(\gamma_2 - \nu_2)^+| + rC(x_1, p).
\end{aligned}
$$

Hence for all $\nu_2 \in \mathcal{M}(X_2)$ we have

$$
\frac{1}{r}[\varphi_1(x_1) - a_1(1-r)^+ - b_1(r-1)^+] \leq \frac{1}{r}\left[ a_2|(\nu_2 - \gamma_2)^+| + b_2|(\gamma_2 - \nu_2)^+| - \int_{X_2} \varphi_2 d\nu_2 \right] + C(x_1, p),
$$

We define $A := \varphi_2^{-1}([-b_2, a_2])$ and $B := \varphi_2^{-1}(-\infty, -b_2) = X_2 \setminus A$, and the Borel bounded function $f : X_2 \to [0, 1]$ by

$$
f(x) := \begin{cases} 1 & \text{if } x \in A, \\ 0 & \text{otherwise}. \end{cases}
$$

Put $\nu_2 = f\gamma_2$. As $J_2(\varphi_2(x)) = \begin{cases} \varphi_2(x) & \text{if } x \in A, \\ -b_2 & \text{if } x \in B, \end{cases}$ we get that

$$
\begin{aligned}
a_2|(\nu_2 - \gamma_2)^+| + b_2|(\gamma_2 - \nu_2)^+| - \int_{X_2} \varphi_2 d\nu_2 &= b_2 \int_{X_2} (1-f) d\gamma_2 - \int_{X_2} \varphi_2 f d\gamma_2 \\
&= b_2 \int_B d\gamma_2 - \int_A \varphi_2 d\gamma_2 \\
&= \int_{X_2} -J_2(\varphi_2) d\gamma_2.
\end{aligned}
$$

Hence for every $r > 0$ we have

$$
\begin{aligned}
\frac{1}{r}[\varphi_1(x_1) - a_1(1-r)^+ - b_1(r-1)^+] &\leq -\frac{1}{r} \int_{X_2} J_2(\varphi_2) d\gamma_2 + C(x_1, p) \\
&= C(x_1, p) - p(J_2(\varphi_2)).
\end{aligned}
$$

Therefore, for every $x_1 \in X_1$, $p \in \mathcal{P}(X_2)$, we get

$$
J_1(\varphi_1(x_1)) = \sup_{r > 0} \frac{\varphi_1(x_1) - a_1(1-r)^+ - b_1(r-1)^+}{r} \leq C(x_1, p) - p(J_2(\varphi_2)).
$$

This implies that $\Phi_E \subset \Phi_J$. Hence $\Phi_E = \Phi_J$.

Now we are ready to prove our duality formula. Moreover, by Lemma A.5 one has $\mathrm{ET}^{\mathbf{k}_1, \mathbf{k}_2}$ is convex and lower semi-continuous. Hence, applying [27, Proposition 3.1, page 14 and Proposition 4.1, page 18] we get that $(\mathrm{ET}^{\mathbf{k}_1, \mathbf{k}_2})^{**} = \mathrm{ET}^{\mathbf{k}_1, \mathbf{k}_2}$. Therefore, applying Lemma A.4, we get

$$
\begin{aligned}
\mathrm{ET}^{\mathbf{k}_1, \mathbf{k}_2}(\mu_1, \mu_2) &= \sup_{(\varphi_1, \varphi_2) \in C_b(X_1) \times C_b(X_2)} \left\{ \sum_{i=1}^{2} \int_{X_i} \varphi_i(x_i) d\mu_i(x_i) - (\mathrm{ET}^{\mathbf{k}_1, \mathbf{k}_2})^*(\varphi_1, \varphi_2) \right\} \\
&= \sup_{(\varphi_1, \varphi_2) \in \Phi_E} \sum_{i=1}^{2} \int_{X_i} \varphi_i(x_i) d\mu_i(x_i) \\
&= \sup_{(\varphi_1, \varphi_2) \in \Phi_J} \sum_{i=1}^{2} \int_{X_i} \varphi_i(x_i) d\mu_i(x_i) \\
&= \sup_{(\varphi_1, \varphi_2) \in \Phi_I} \sum_{i=1}^{2} \int_{X_i} I_i(\varphi_i(x_i)) d\mu_i(x_i).
\end{aligned}
$$

$\square$

## A.4 Proof of Corollary 3.1

*Proof of Corollary 3.1.* We define $C$ : $X_1 \times \mathcal{P}(X_2) \to (-\infty, +\infty]$ by $C(x_1, p) := k \int_{X_2} c(x_1, x_2) dp(x_2)$ for every $x_1 \in X_1$, $p \in \mathcal{P}(X_2)$. Then $C$ is bounded from below, $\frac{b_1}{2} + \frac{b_2}{2} + \inf C > 0$, and $C(x_1, \cdot)$ is convex for every $x_1 \in X_1$. Applying [22, Lemma 14], we get that $C$ is lower semicontinuous. From Example A.1-A.3, we also have that $C$ has property (T).

Now we prove that $\Phi_I = \Lambda_I$. For every $(\varphi_1, \varphi_2) \in \Phi_I$, $x_1 \in X_1, x_2 \in X_2$ we have

$$\varphi_1(x_1) + \varphi_2(x_2) = \varphi_1(x_1) + \delta_{x_2}(\varphi_2)$$
$$= \int_{X_2} (\varphi_1(x_1) + \varphi_2(y)) d\delta_{x_2}(y)$$
$$\leq \int_{X_2} k \cdot c(x_1, y) d\delta_{x_2}(y)$$
$$= k \cdot c(x_1, x_2).$$

Hence $\Phi_I \subset \Lambda_I$. On the other hand, for every $(\varphi_1, \varphi_2) \in \Lambda_I$, $x_1 \in X_1, p \in \mathcal{P}(X_2)$, we have

$$\varphi_1(x_1) + p(\varphi_2) = \int_{X_2} (\varphi_1(x_1) + \varphi_2(y)) dp(y)$$
$$\leq \int_{X_2} k \cdot c(x_1, y) dp(y)$$
$$= C(x_1, p).$$

Hence $\Lambda_I \subset \Phi_I$, and therefore $\Phi_I = \Lambda_I$. Similarly, we also have $\Phi_J = \Lambda_J$. Applying Theorem 3.2, we get the result. $\square$

## A.5 Proof of Theorem 3.3

*Proof of Theorem 3.3.* Applying Corollary 3.1 we have

$$\mathcal{E}^{\mathbf{k}_1, \mathbf{k}_2, k}(\mu_1, \mu_2) = \sup \left\{ \int_{X_1} I_1(\varphi_1(x)) d\mu_1(x) + \int_{X_2} I_2(\varphi_2(y)) d\mu_2(y) : \varphi_i \in C_b(X_i), \right.$$
$$\left. \varphi_1(x) \geq -b_1, \varphi_2(y) \geq -b_2, \varphi_1(x) + \varphi_2(y) \leq k \cdot c(x, y), \forall x \in X_1, y \in X_2 \right\}.$$

As $\mu_2 = \sum_{i=1}^{M} m_i \delta_{y_i}$, we get that

$$\mathcal{E}^{\mathbf{k}_1, \mathbf{k}_2, k}(\mu_1, \mu_2) = \sup \left\{ \int_{X_1} I_1(\varphi_1(x)) d\mu(x) + \sum_{i=1}^{M} m_i I_2(w_i) : \varphi_1 \in C_b(X_1), \varphi_1(x) \geq -b_1, \right.$$
$$\left. w_i \geq -b_2, \varphi_1(x) + w_i \leq k \cdot c(x, y_i), \forall x \in X_1, i = 1, \ldots, M \right\}.$$

As $c$ is lower semicontinuous and satisfies *radial* property (see Definition 3.2) with respect to $\mu_1$, we obtain that $\{C_1(w), \ldots, C_M(w), R\}$ is a $\mu_1$-measurable partition of $X_1$. Given $n \in \mathbb{N}$ and $w = (w_1, \ldots, w_M) \in [-b_2, +\infty)^M$, we define the map $c_n^w : X_1 \to (-\infty, +\infty]$ by $c_n^w(x) = \min\{n, c^w(x)\}\}$. As $c$ is measurable, so are $c^w$ and $c_n^w$. Since $\{c_n^w\}_{n \in \mathbb{N}}$ increases, converges pointwise to $c^w$, $I_1$ is continuous and increasing on $[-b_1, +\infty)$, and $\lim_{s \to \infty} I_1(s) = a_1$, we get that

$$\lim_{n \to \infty} \int_{X_1} I_1(c_n^w(x)) d\mu_1(x) = \lim_{n \to \infty} \left[ \sum_{i=1}^{M} \int_{C_i(w)} I_1(c_n^w(x)) d\mu_1(x) + \int_R I_1(c_n^w(x)) d\mu_1(x) \right]$$
$$= \sum_{i=1}^{M} \int_{C_i(w)} I_1(-b_1 \vee [k \cdot c(x, y_i) - w_i])(x)) d\mu_1(x) + a_1 \mu_1(R).$$

Therefore,

$$\mathcal{E}^{\mathbf{k_1},\mathbf{k_2},k}(\mu_1,\mu_2) = \sup\left\{\mathcal{G}(w) : w = (w_1,\ldots w_M) \in [-b_2,+\infty)^M\right\}.$$

$\square$

## A.6   Proof of Theorem 3.4

*Proof of Theorem 3.4.* Put $B_i = X_1 \times \{y_i\}, i = 1,\ldots,M$. Then $\{B_i : i = 1,\ldots,M\}$ is a measurable partition of $X_1 \times X_2$ and hence we can write $\gamma = \sum_{i=1}^{M} \eta_i \otimes \delta_{y_i}$ for $\eta_i \in \mathcal{M}(X_1), i = 1,\ldots,M$. Put $f = \dfrac{d\gamma_1}{d\mu_1}$. As $\mathcal{E}^{\mathbf{k_1},\mathbf{k_2},k}(\gamma|\mu_1,\mu_2) < \infty$ we get $\int_{X_1 \times X_2} cd\gamma < \infty$ and hence $\gamma_1(R) = 0$. Therefore $\int_R f d\mu_1 = 0$, i.e., $f(x) = 0$ for $\mu_1$-a.e. $x \in R$. The Lebesgue decomposition of $\gamma_2 = \pi_{2\#}\gamma$ with respect to $\mu_2$ is $\gamma_2 = \sum_{i \in S_1} |\eta_i|\delta_{x_i} + \gamma_2^{\perp}$, where $\gamma_2^{\perp} = \sum_{i \in S_2} |\eta_i|\delta_{x_i}$. Here $\pi_{i\#}\gamma$ refers to the $i$th marginal of $\gamma$. We have

$$
\begin{aligned}
\mathcal{E}^{\mathbf{k_1},\mathbf{k_2},k}(\gamma|\mu_1,\mu_2) =& k\int_{X_1 \times \{y_1,\ldots,y_M\}} cd\gamma + \int_{X_1} [a_1(1-f(x))^+ + b_1(f(x)-1)^+]d\mu_1(x) + \\
& + b_1\gamma_1^{\perp}(X_1) + \sum_{i \in S_1}\left[a_2\left(1 - \frac{|\eta_i|}{m_i}\right)^+ + b_2\left(\frac{|\eta_i|}{m_i} - 1\right)^+\right]m_i + b_2\sum_{i \in S_2}|\eta_i| \\
=& \sum_{i=1}^{M} k\int_{X_1 \setminus R} c(x,y_i)d\eta_i(x) + \int_{X_1 \setminus R} [a_1(1-f(x))^+ + b_1(f(x)-1)^+]d\mu_1(x) + \\
& + a_1\mu(R) + b_1\gamma_1^{\perp}(X_1) + \sum_{i \in S_1}\left[a_2\left(1 - \frac{|\eta_i|}{m_i}\right)^+ + b_2\left(\frac{|\eta_i|}{m_i} - 1\right)^+\right]m_i + b_2\sum_{i \in S_2}|\eta_i|.
\end{aligned}
$$

On the other hand, for $i = 1,2$, we have

$$a_i(1-s)^+ + b_i(s-1)^+ - I_i(\varphi) \geq \begin{cases} -sa_i & \text{if } \varphi > a_i, \\ -s\varphi & \text{if } -b_i \leq \varphi \leq a_i, \\ \infty & \text{otherwise.} \end{cases}$$

Hence $a_i(1-s)^+ + b_i(s-1)^+ - I_i(\varphi) \geq -s\varphi$ for all $s, \varphi$. The inequality is an equality if and only if $-b_i \leq \varphi$ and $s = 1$. Therefore, we get that

$$
\begin{aligned}
\mathcal{E}^{\mathbf{k_1},\mathbf{k_2},k}(\gamma|\mu_1,\mu_2) - \mathcal{G}(w) =& k\sum_{i=1}^{M}\int_{X_1 \setminus R} c(x,y_i)d\eta_i(x) + b_1\gamma_1^{\perp}(X_1) + b_2\sum_{i \in S_2}|\eta_i| \\
& + \int_{X_1 \setminus R}\left[a_1(1-f(x))^+ + b_1(f(x)-1)^+ - I_1(c^w(x))\right]d\mu_1(x) \\
& + \sum_{i \in S_1}\left[a_2\left(1 - \frac{|\eta_i|}{m_i}\right)^+ + b_2\left(\frac{|\eta_i|}{m_i} - 1\right)^+ - I_2(w_i)\right]m_i \\
\geq& \sum_{i=1}^{M}\int_{X_1 \setminus R} c(x,y_i)d\eta_i(x) + b_1\gamma_1^{\perp}(X_1) + b_2\sum_{i \in S_2}|\eta_i| \\
& - \int_{X_1 \setminus R} c^w(x)f d\mu_1(x) - \sum_{i \in S_1}|\eta_i|w_i \\
=& \sum_{i=1}^{M}\int_{X_1 \setminus R}\left[kc(x,y_i) - w_i - c^w(x)\right]d\eta_i(x) + \int_{X_1 \setminus R} c^w(x)d\gamma_1^{\perp}(x) \\
& + b_1\gamma_1^{\perp}(X_1 \setminus R) + \sum_{i \in S_2}|\eta_i|w_i + b_2\sum_{i \in S_2}|\eta_i| \\
=& \sum_{i=1}^{M}\int_{C_i(w)}\left[kc(x,y_i) - w_i - c^w(x)\right]d\eta_i(x) + \int_{X_1 \setminus R}[c^w(x) + b_1]d\gamma_1^{\perp}(x)
\end{aligned}
$$

$$+ \sum_{i \in S_2} |\eta_i|(w_i + b_2)$$

$$\geq 0.$$

The first inequality is an equality if and only if $\dfrac{d\gamma_1}{d\mu_1}(x) = f(x) = 1$ for $\mu_1$-a.e. $x \in X_1 \setminus R$ and $\dfrac{|\eta_i|}{m_i} = 1$ for all $i \in S_1$. The second inequality becomes equality if and only if $\mathrm{supp}\eta_i \subset C_i(w)$ and hence $\eta_i = \gamma_1 \llcorner C_i(w)$ for $i = 1, \ldots, M$, $c^w(x) = -b_1$ for $\gamma_1^{\perp}$-a.e. $x \in X_1 \setminus R$, and $w_i = -b_2$ for every $i \in S_2$. Therefore, we get that $\mathcal{E}^{\mathbf{k}_1, \mathbf{k}_2, k}(\gamma|\mu_1, \mu_2) - \mathcal{G}(w) = 0$ if and only if conditions (7-10) hold. $\qquad\square$

## A.7   Proof of Proposition 3.1

*Proof of Proposition 3.1.* By Definition 3.3, we have $c^w(x) = +\infty$ for any $x \in R$. By (3), for any $x \in R$, we have $I_1(c^w(x)) = I_1(+\infty) = a_1$. Thus, $a_1\mu_1(R) = \int_R I_1(c^w(x))d\mu_1(x)$. Combining this result with (6), we obtain

$$\mathcal{G}(w) = \int_{X_1} I_1(c^w(x))d\mu_1(x) + \sum_{i=1}^{M} I_2(w_i)m_i.$$

By definition in (3), $I_1$ is differentiable on $(-b_1, a_1) \cup (a_1, +\infty)$, such that $\partial_x I_1(x) = 1$ on $(-b_1, a_1)$ and $\partial_x I_1(x) = 0$ on $(a_1, +\infty)$. Thus, for any $x \in C_i(w)$, it holds $I_1(c^w(x)) = I_1((-b_1 \vee k \cdot c(x, y_i) - w_i)$, and we can derive the derivatives

$$\partial_{w_i} I_1(c^w(x)) = \begin{cases} 0, & k \cdot c(x, y_i) - w_i \in (-\infty, b_1), \\ [-1, 0], & k \cdot c(x, y_i) - w_i \in \{b_1\}, \\ -1, & k \cdot c(x, y_i) - w_i \in (-b_1, a_1), \\ [-1, 0], & k \cdot c(x, y_i) - w_i \in \{a_1\}, \\ 0, & k \cdot c(x, y_i) - w_i \in (a_1, +\infty). \end{cases}$$

On the other hand, for any $x \in C_i(w)$, we have $\partial_{w_j} I_1(c^w(x)) = 0$, for any $j \neq i$. Therefore, by Lemma 3.8 of [15], *radial* property (see Definition 3.2) of cost function $c$, and the absolutely continuity of $\mu_1$ with respect to Lebesgue measure, then

$$\partial_{w_i} \int_{\Omega} I_1(c^w(x))d\mu_1(x) = -\mu_1(C_i(w) \cap \{x : k \cdot c(x, y_i) - w_i \in [-b_1, a_1]\}). \qquad (17)$$

Similarly, by definition in (3), we know

$$\partial_{w_i} I_2(w_i) = \begin{cases} 1, & w_i \in (-b_2, a_2), \\ [0, 1], & w_i \in \{a_2\}, \\ 0, & w_i \in (a_2, +\infty). \end{cases}$$

Then the (sub-)gradient can be obtained as presented.

$\qquad\square$

## A.8   Proof of Theorem 3.5

Equations (7-10) show that continuous mass rearranges only within Laguerre cells, while the residual mass collapses into the remainder set $R$. This insight later underpins our quantization and pruning strategies. Plugging the optimality conditions of Theorem 3.4 back yields the following explicit primal form, facilitating comparison with classical semi-discrete OT.

**Corollary A.1 (Primal tessellation formulation).** *If there exists optimizers for* (2) *and* (5)*, then*

$$\mathcal{E}^{\mathbf{k}_1, \mathbf{k}_2, k}(\mu_1, \mu_2) = \min\Bigg\{ k \sum_{i=1}^{M} \int_{C_i(w)} c(x, y_i)d\gamma_1(x) + a_1|(\mu_1 - \gamma_1)^+| + b_1|(\gamma_1 - \mu_1)^+| +$$

$$(18)$$

$$\sum_{i \in S_1}\left[a_2(1 - \frac{|\eta_i|}{m_i})^+ + b_2(\frac{|\eta_i|}{m_i} - 1)^+\right]m_i + b_2 \sum_{i \in S_2}|\eta_i| \,\Bigg|\, w \in [-b_2, +\infty)^M, \gamma_1 \in \mathcal{M}(X_1), \gamma_1 \llcorner R = 0\Bigg\},$$

*where* $\eta_i = \gamma_1 \llcorner C_i(w), i = 1, \cdots, M$. *If* $\gamma$ *and* $w$ *are optimizers for* (2) *and* (5)*, then the first margin* $\gamma_1$ *and* $w$ *are optimizers for* (18)*. Conversely, if* $\gamma_1$ *and* $w$ *are optimizers for* (18)*, then* $\gamma$ *defined by* (7) *is the optimizer for* (2)*.*

*Proof.* For any choice of $w \in [-b_2, +\infty)^M$, $\gamma_1 \in \mathcal{M}(X_1)$, and $\gamma_1 \llcorner R = 0$, define $\gamma = \sum_{i=1}^{M} \gamma_1 \llcorner C_i(w) \otimes \delta_{y_i} \in \mathcal{M}(X_1 \times X_2)$. The right-hand side of (18) can be written in the same form as (2) where the admissible set for $\gamma$ is restricted to a subset such that $\gamma$ takes the form $\gamma = \sum_{i=1}^{M} \gamma_1 \llcorner C_i(w) \otimes \delta_{y_i}$. Therefore, the right-hand side of (18) is no less than $\mathcal{E}^{\mathbf{k}_1, \mathbf{k}_2, k}(\mu_1, \mu_2)$. On the other hand, by Theorem 3.4, if $\gamma^* = \sum_{i=1}^{M} \eta_i \otimes \delta_{y_i}$ and $w^* \in [-b_2, +\infty)^M$ are taken to be the optimizers for (2) and (5), respectively, then they should satisfy (7-10), which leads to $\eta_i = \gamma_1^* \llcorner C_i(w), i = 1, \cdots, M$, and $\gamma_1^* \llcorner R = 0$, where $\gamma_1^*$ is the first margin of $\gamma^*$. In this case, the right-hand side of (18) equals $\mathcal{E}^{\mathbf{k}_1, \mathbf{k}_2, k}(\mu_1, \mu_2)$, and $\gamma_1^*$ and $w^*$ are the optimizers for (18).

Conversely, if $\gamma_1^*$ and $w^*$ are optimizers for (18), then $\gamma$ defined by (7) in terms of $\gamma_1^*$ and $w^*$ is the optimizer for (2). $\qquad\square$

*Proof of Theorem 3.5.* Consider any fixed $\{y_1, \cdots, y_M\} \subset X_2$. For any constant $a \in \mathbb{R}$, let $C_i(a) := C_i((a, \cdots, a))$, where $(a, \cdots, a)$ is an $M$-dimensional vector. For every $w \in [-b_2, +\infty)^M$, we define

$$\mathcal{G}(w, m_1, \ldots, m_M) := \int_{X_1 \setminus R} I_1(c^w(x)) d\mu_1(x) + a_1 \mu_1(R) + \sum_{i=1}^{M} I_2(w_i) m_i,$$

where $c^w : X_1 \to (-\infty, +\infty]$ is defined by $c^w(x) = \min\{k \cdot c(x, y_i) - w_i : i = 1, \ldots, M\}\} \vee -b_1$.

From Theorem 3.3 we get that

$$\mathcal{E}^{\mathbf{k}_1, \mathbf{k}_2, k}(\mu_1, \mu_2) \geq \mathcal{G}(\mathbf{0}, m_1, \ldots, m_M).$$

As $I_2(0) = 0$ we get that $\mathcal{G}(\mathbf{0}, m_1, \ldots, m_M) = \int_{X_1 \setminus R} I_1(c^{\mathbf{0}}(x)) d\mu_1(x) + a_1 \mu_1(R) =: \mathcal{G}(\mathbf{0})$ does not depend on $m_1, \ldots, m_M$.

Let $\mu_2^* = \sum_{i=1}^{M} m_i^* \delta_{y_i}$. We define the measurable map $f : X_1 \to \mathbb{R}$, defined by $f(x) = 1$ for $\mu_1$-a.e $x \in X_1 \setminus R$ and $f(x) = 0$ for $\mu_1$-a.e $x \in R$. Define $\eta^* := f \mu_1 \in \mathcal{M}(X_1)$, $\eta_i^* := \eta^* \llcorner C_i(\mathbf{0})$, and $\gamma^* := \sum_{i=1}^{M} \eta^* \llcorner C_i(\mathbf{0}) \otimes \delta_{\mathbf{y_i}}$. Then the Lebesgue decomposition of $\gamma_1^*$ with respect to $\mu_1$ is $\gamma_1^* = \sum_{i=1}^{M} \eta_i^* = f \mu_1$. Optimality conditions (10) is fulfilled. By Theorem 3.4, it holds that $\mathcal{E}^{\mathbf{k}_1, \mathbf{k}_2, k}(\mu_1, \mu_2^*) = \mathcal{G}(\mathbf{0})$. Hence $Q(y_1, \cdots, y_M) = \mathcal{G}(\mathbf{0})$, i.e.

$$Q(y_1, \cdots, y_M) = \sum_{i=1}^{M} \int_{C_i(\mathbf{0})} I_1(-b_1 \vee k \cdot c(x, y_i)) d\mu_1(x) + a_1 \mu_1(R).$$

$\qquad\square$

## A.9 Proof of Proposition 3.2

*Proof of Proposition 3.2.* Note that

$$Q(y_1, \cdots, y_M) = \int_{X_1} I_1(c^{\mathbf{0}}(x)) d\mu_1(x).$$

By definition in (3), we know

$$\partial_x I_1(x) = \begin{cases} 1, & x \in (-b_1, a_1), \\ [0, 1], & x \in \{a_1\}, \\ 0, & x \in (a_1, +\infty). \end{cases}$$

Fix $x \in C_i(\mathbf{0})$, we have $I_1(c^{\mathbf{0}}(x)) = I_1((-b_1 \vee k \cdot c(x, y_i))$, and its derivative is obtained by chain rule,

$$\nabla_{y_i} I_1(c^{\mathbf{0}}(x)) = \begin{cases} 0, & k \cdot c(x, y_i) \in (-\infty, b_1), \\ k \nabla_{y_i} c(x, y_i), & k \cdot c(x, y_i) \in (-b_1, a_1), \\ 0, & k \cdot c(x, y_i) \in (a_1, +\infty). \end{cases}$$

Besides, $\nabla_{y_j} I_1(c^{\mathbf{0}}(x)) = 0$ for any $j \neq i$. Therefore,

$$\nabla_{y_i} Q(y_1, \cdots, y_M) = \int_{C_i(\mathbf{0})} k \nabla_{y_i} c(x, y_i) \mathbf{1}_{\{x : k \cdot c(x, y_i) \in (-b_1, a_1)\}} d\mu_1(x)$$

$$= k \int_{C_i(\mathbf{0}) \cap \{\mathbf{x} : \mathbf{k} \cdot \mathbf{c}(\mathbf{x}, \mathbf{y_i}) \in (-\mathbf{b_1}, \mathbf{a_1})\}} \nabla_{y_i} c(x, y_i) d\mu_1(x).$$

$\square$

