# OpenReview forum: "Unbalanced Optimal Total Variation Transport: A Theoretical Approach to Spatial Resource Allocation Problems"
_NeurIPS.cc/2025/Conference — NeurIPS 2025 poster_

### Official Review · Reviewer_8tzK · 2025-06-30

**Clarity:** 3
**Significance:** 2
**Originality:** 3
**Rating:** 4
**Confidence:** 3

**Summary:**

The paper investigates an unbalanced, barycentric (i.e., weak), total variation-regularized optimal transport formulation. The authors show the existence of a solution for the considered problem and derive its dual form. They then discuss the corresponding semi-discrete setting, showing that it is equivalent to an optimization problem on Laguerre diagrams and providing its optimality conditions. They further extend this setting into optimizing over the space of discrete measures (i.e., weights and support points), which is inspired by the problem of spatial resource allocation, and provide an alternating scheme for solving it. Experimental results regarding those semi-discrete settings are carried out on synthetic data.

**Questions:**

- What is the importance of considering weak optimal transport in the proposed problem? I do not see its relevance to the allocation problem. Is it for the technicality of TV regularizers?
- The paper mentions entropic regularization several times in Section 2, but the problem of interest is not entropy-regularized, as well as the formulation written in Line 101.
- Typo: $\phi$ instead of $\varphi$ in Line 191

**Ethical Concerns:**

["NO or VERY MINOR ethics concerns only"]

**Final Justification:**

I maintain my score of 4, indicating a slight lean toward acceptance, based on the following:

Pros: The paper presents a well-rounded contribution. The formulation is novel, supported by sufficient theoretical results and practical algorithms, and further validated through a real-world application.

Cons: The theoretical and algorithmic techniques show limited novelty, and the empirical evaluation remains insufficiently thorough.

In their response, the authors provided additional details on the resource allocation problem, which I believe will improve the paper’s overall quality. Comments on the use of the weak OT formulation (which is also noted by other reviewers) would further strengthen its clarity. However, the primary concerns regarding novelty and empirical depth remain.

**Limitations:**

Yes

**Quality:**

3

**Strengths And Weaknesses:**

Strengths:
- The paper is well-written and well-presented.
- The proposed problem is new, and so are the relevant results.

Weaknesses:
- The motivation and the final problem of the paper aim at the practical problem of spatial resource allocation, but there is little discussion about the related work on this topic, as well as the experiment results are only on synthetic data and without comparison to any other method.
- The alternating scheme might converge only to a local minima, as the loss seems to be non-convex.

---

> ### Author Rebuttal · Authors · 2025-07-31
>
> We sincerely appreciate your positive feedback on our work, as well as your thoughtful and constructive comments. Please find our detailed responses to each of your comments and questions below.
>
> **[1]** What is the importance of considering weak optimal transport in the proposed problem?
>
> ## R1
> Thank you for giving us the chance to clarify. Weak OT (WOT) is a genuine generalization of classical OT. In classical OT, each source point $x$ is coupled with a Dirac measure at a single target point, i.e., the coupling $\pi(\cdot\mid x)$ is deterministic. In contrast, WOT allows $\pi(\cdot\mid x)$ to be any probability measure, enabling stochastic transport from each source location. As a result, the classical OT feasible set is a strict subset of the WOT feasible set. When the cost function $c(x,\cdot)$ is convex, WOT serves as a relaxation of the classical problem and can only decrease the objective value.
>
> From a theoretical perspective, particularly for a theory-track paper, our aim is to develop as general a framework as possible and push the theory to the limit of our method. The weak formulation allows us to extend the theory to its broadest scope while naturally including classical OT as a special case.
>
> Beyond its theoretical generality, the weak formulation offers practical advantages and has attracted independent interest in several areas. By changing the coupling constraints, WOT improves robustness to sampling noise and enhances stability in empirical settings where exact couplings may not exist or may be unstable due to data perturbations (see, e.g., [1,2]). While a detailed exploration of such applications is beyond the current scope and left for future work, we believe that the inclusion of the weak formulation is not merely a technical generalization, but rather reflects the theoretical flexibility and broader potential of our proposed approach.
>
> **[2]** The paper mentions entropic regularization several times in Section 2, but the problem of interest is not entropy-regularized, as well as the formulation written in Line 101.
>
> ## R2
> Thanks for your clarification question. The references to entropic regularization in Section 2 are included solely to position our work within the existing semi-discrete unbalanced OT literature, where Kullback–Leibler or other entropy-based penalties are commonly used.
>
> The main contribution of our paper lies in the development of a comprehensive theoretical framework for a new class of unbalanced weak OT problems incorporating total variation (TV) penalties, along with an illustration of its applicability to spatial resource allocation. Building on the work [3], we take one step forward, and to the best of our knowledge, this is the first theoretical study to address weak OT problems with TV penalties in the unbalanced setting. Importantly, the motivation for incorporating TV penalties is both technical and practical. Technically, TV regularization is well-suited to settings with mismatched masses, a common feature in real-world problems where supply–demand imbalance is the norm. Practically, the TV penalty is directly interpretable: If supply exceeds demand, a proportional overage cost is incurred, reflecting the cost of excess supply (e.g., idle resources, inventory holding); When demand exceeds supply, a shortage cost is applied, corresponding to the opportunity loss from unmet demand, such as forgone revenue. This structure provides an intuitive and economically meaningful way to quantify mismatch costs, particularly relevant for our focus on resource allocation applications, as emphasized in the paper's title. In contrast, entropy-based regularization, while analytically convenient, offers less interpretability in such contexts.
>
> Adopting a TV regularization term fundamentally changes the problem and raises new challenges: the objective becomes less smooth, loses the strong convexity enjoyed by entropic formulations, and thus precludes the direct use of prior methods developed under those assumptions (e.g., [3]). So we develop totally different theoretical analysis (compared to [3]). As a result, we develop new theoretical tools tailored to this setting. This is precisely why the entropic framework is cited only for contextual comparison and is not employed in our formulation.
>
>
> **[3]** Typo: $\phi$ instead of $\varphi$ in Line 191.
> ## R3
> Thank you very much for your careful reading and for diligently pointing out this typo. We have corrected it in the revised version of the manuscript.
>
>
> **[4]** The final problem of the paper aims at the practical problem of spatial resource allocation, but there is little discussion about the related work on this topic, and the experiment results are only on synthetic data and without comparison to any other method.
> ## R4
> Thank you for your thoughtful comment. To the best of our knowledge, there is currently no existing work on semi-discrete unbalanced OT that optimizes a TV-regularized objective. Existing algorithms in the literature are typically built around entropic (e.g., KL-divergence) regularization, and their convergence guarantees rely heavily on the strong convexity and smoothness properties specific to that setting. As such, these methods are not directly adaptable to the TV-penalized formulation we consider.
>
> In contrast, our work introduces a TV-based regularization, motivated by both theoretical considerations and practical relevance. The TV penalty is intuitive and interpretable, directly measuring the number of unmet or excessive units in a supply–demand setting. For instance, DiDi—a leading ride-sourcing platform in China—has employed TV-style loss metrics to evaluate operational inefficiencies due to idle supply or unmet demand (see [4]). While entropy-based regularization is widely used for computational convenience (see [1–4,14]), it lacks the same level of interpretability in operational contexts. Despite its practical appeal, TV regularization remains underexplored in the theoretical OT literature, making our proposed framework a natural and meaningful extension.
>
> From an application standpoint, several works relate to the broader theme of resource allocation with imbalance (see [5–11]). Among them, the most relevant is [11], which addresses order dispatching in ride-sourcing platforms. That model can be viewed as a special case of our framework, incorporating application-specific constraints such as fixed supply-side marginals to represent limited driver availability.
>
> Regarding experimental results: as noted above, there is currently no available algorithm for semi-discrete unbalanced OT with TV regularization. The closest prior work (e.g., [12–15]) incorporates imbalance through entropic penalties, not TV. Hence, direct numerical comparisons are not meaningful, as existing methods do not address the same problem formulation.
>
> Finally, we emphasize that this is a theory-track submission, and our main contribution is the development of a general theoretical framework. The optimization algorithm included in the semi-discrete case is intended as an illustrative application, rather than the central focus. Including extensive algorithmic benchmarks would shift the emphasis away from the core theoretical contributions and exceed the page limits. We appreciate your understanding on this point.
>
>
> **[5]** The alternating scheme might converge only to a local minima, as the loss seems to be non-convex.
>
> ## R5
> Regarding the convergence of our algorithm, we kindly ask you to refer to our response to Reviewer ZhcG (R2), as we are constrained by the character limit imposed by the conference system. We greatly appreciate your understanding.
>
>
> Thank you once again for your insightful and constructive feedback on our manuscript. We hope you find our revision satisfactory.
>
> [1] Julio Backhoff-Veraguas and Gudmund Pammer. Applications of weak transport theory. Bernoulli, 28(1):370–394, 2022.
>
> [2] Jean Feydy, Thibault Séjourné, François-Xavier Vialard, Shun-ichi Amari, Alain Trouvé, and Gabriel Peyré. Interpolating between optimal transport and mmd using sinkhorn divergences. AISTATS, pages 2681–2690. PMLR, 2019.
>
> [3] Matthias Liero, Alexander Mielke, and Giuseppe Savaré. Optimal entropy-transport problems and a new hellinger–kantorovich distance between positive measures. Inventiones Mathematicae, 211(3):969–1117, 2018.
>
> [4] Fan Zhou, Shikai Luo, Xiaohu Qie, Jieping Ye, and Hongtu Zhu. Graph-based equilibrium metrics for dynamic supply–demand systems with applications to ride-sourcing platforms. JASA, 116(536):1688–1699, 2021.
>
> [5] Benjaafar, Saif, et al. Dynamic inventory repositioning in on-demand rental networks. MS 68.11 (2022): 7861-7878.
>
> [6] Besbes, Omar, Francisco Castro, and Ilan Lobel. Surge pricing and its spatial supply response. MS 67.3 (2021): 1350-1367.
>
> [7] Besbes, Omar, Francisco Castro, and Ilan Lobel. Spatial capacity planning. OR 70.2 (2022): 1271-1291.
>
> [8] Bimpikis, Kostas, Ozan Candogan, and Daniela Saban. Spatial pricing in ride-sharing networks. OR 67.3 (2019): 744-769.
>
> [9] Hosseini, Mahsa, Joseph Milner, and Gonzalo Romero. Dynamic relocations in car-sharing networks. OR 73.4 (2025): 2010-2025.
>
> [10] Gunnar Carlsson, John, Xiaoshan Peng, and Ilya O. Ryzhov. Demand equilibria in spatial service systems. MSOM 26.6 (2024): 2305-2321.
>
> [11] Zhou, Fan, et al. Graph-based equilibrium metrics for dynamic supply–demand systems with applications to ride-sourcing platforms. JASA 116.536 (2021): 1688-1699.
>
> [12] Chizat, Lenaic, et al. Scaling algorithms for unbalanced optimal transport problems. Mathematics of computation 87.314 (2018): 2563-2609.
>
> [13] Scetbon, Meyer, et al. Unbalanced low-rank optimal transport solvers. NeurIPS 36 (2023): 52312-52325.
>
> [14] Pooladian, Aram-Alexandre, Vincent Divol, and Jonathan Niles-Weed. Minimax estimation of discontinuous optimal transport maps: The semi-discrete case. ICML, 2023.
>
> [15] Bonet, Clément, et al. Slicing Unbalanced Optimal Transport. TMLR (2024).

---

> > ### Comment · Reviewer_8tzK · 2025-08-06
> > **Rebuttal Response**
> >
> > I thank the authors for their detailed responses, which I find satisfactory. I will maintain my current score, leaning toward acceptance.

---

> > > ### Author Response · Authors · 2025-08-09
> > > **Follow-up on Rebuttal and Gratitude for Your Kind Comments**
> > >
> > > Dear Reviewer 8tzK,
> > >
> > > We truly appreciate the time and effort you devoted to reading our rebuttal and for expressing a positive assessment of our work. It is encouraging to know that our responses have satisfactorily addressed your points.
> > >
> > > Should there be any remaining questions or concerns, please let us know and we will gladly provide further clarification. If everything is clear, we would be thankful if you might consider adjusting your score upward in view of the additional explanations and results included in our rebuttal.
> > >
> > > Thank you once again for your constructive review and for contributing to a fair and balanced evaluation process.
> > >
> > > Best regards,
> > > The Authors

---

### Official Review · Reviewer_z3hw · 2025-07-01

**Clarity:** 4
**Significance:** 3
**Originality:** 3
**Rating:** 5
**Confidence:** 1

**Summary:**

The paper proposes a novel semi-discrete, unbalanced weak optimal transport model with total variation penalties for spatial resource allocation problems. It provides theoretical analysis and extends classical transport frameworks to non-smooth, TV-regularized settings. Experiments demonstrate promising results on small-scale problems, but scalability and hyperparameter sensitivity warrant further investigation.

**Questions:**

1. The experimental evaluation is limited to small-scale applications. Have the authors considered testing their method on larger-scale datasets or more complex scenarios to further assess its scalability, performance, and convergence properties?
2. The paper involves several hyperparameters in the proposed method. Could the authors elaborate on the process of hyperparameter selection and provide an analysis of how sensitive the algorithm's performance is to these choices?

**Ethical Concerns:**

["NO or VERY MINOR ethics concerns only"]

**Final Justification:**

I remain positive on the paper, including the writing, the method and the theoretical results. The submission is not in my area, so I have lowest confidence about my score.

**Limitations:**

yes

**Quality:**

4

**Strengths And Weaknesses:**

**Strength**

1. The authors model a novel semi-discrete, unbalanced weak optimal transport problem with total variation penalties, motivated by practical spatial resource allocation problems.
2. For this problem, the authors conduct theoretical analyses on the existence of optimal solutions and tessellation-based reformulations, and extend classical entropy-regularized transport frameworks to the non-smooth, TV-penalized setting.

**Weakness**

1. The experiments are conducted only on small-scale applications. Although the results are promising, the performance and convergence on larger-scale problems remain to be verified.
2. Based on my previous research experience, I have often found that theoretically sound methods may have limited practical effectiveness, and may require careful tuning of hyperparameters to ensure convergence and satisfactory performance. The experiments in this paper also involve various hyperparameters; more discussion is needed regarding how they are selected and their impact on performance.

---

> ### Author Rebuttal · Authors · 2025-07-31
>
> We sincerely appreciate your positive feedback on our work, as well as your thoughtful and constructive comments. Please find our detailed responses to each of your questions below.
>
> **[1]** The experimental evaluation is limited to small-scale applications. Have the authors considered testing their method on larger-scale datasets or more complex scenarios to further assess its scalability, performance, and convergence properties?
>
> ## R1
> Thanks for your comment. As a theory-track submission, the primary contribution of our work lies in the development of a comprehensive theoretical framework for a new class of unbalanced weak OT problems incorporating total variation (TV) penalties. The optimization routine presented in the (special) semi-discrete setting is therefore intended not as the main focus, but rather as an illustrative application of the proposed framework. That said, we appreciate your suggestion to incorporate more experiments on larger-scale datasets or more complex scenarios to further assess its scalability, performance, and convergence properties. While these are certainly valuable and interesting directions, we have opted to leave them for future work for two reasons: (i) to maintain clarity and focus on the general theoretical framework, and (ii) due to space limitations.
>
> In particular, we also appreciate your concern regarding scalability. As an illustration, we extended the depot-location experiment in Section 4.2 to five larger scales, increasing the number of facilities $m$ and the dimensionality of the optimization variable (from 12 to 192 parameters). We sample \(N=2000\) incident locations from a two‑component Gaussian mixture with means $(0.3d,0.3d)$ and $(0.7d,0.7d)$, and covariance $0.02I_2$. Samples are clipped to $[0,d]^2$ and rescaled to ensure $M_\mu=1$.The number of depots is $m$, with initial positions drawn i.i.d. from the uniform distribution over $[0,d]^2$. TV parameters are set as $a_1=0.3$, $a_2=1$, $b_1=0.1$, and $b_2=0.2$. A decaying step size $0.3/(1+t)^{0.6}$ is used.
>
> We reran the spatial‑allocation test with progressively larger depot sets. The CPU time rises almost linearly, within an acceptable speed. Although this scalability assessment is not the focus of our paper, we believe it demonstrates the practical potential of our theoretical findings and provides a promising direction for future investigation. Since we are unable to upload images or PDF files, we present the results in tabular form.
> | scale (m,d) | (4,1) | (8,2) |(16,3) | (32,3) |(64,4)|
> |-|-|-|-|-|-|
> | numbers of parameter | 12 | 24 | 48 | 96 |192|
> |CPU time |4.9s|7.5s|15.2s|29.6s|65.9s|
>
>
> **[2}** The paper involves several hyperparameters in the proposed method. Could the authors elaborate on the process of hyperparameter selection and provide an analysis of how sensitive the algorithm's performance is to these choices?
>
> ## R2
> Thanks for your insightful question. While our model does involve several parameters, each of them reflects practical considerations and carries clear interpretability in real-world applications. For instance, if we interpret $\mu_1$ as the supply distribution and $\mu_2$ as the demand distribution, the parameters in the cost objective (Equation (2)) have the following meanings: $a_1$ measures the unit cost excess supply (e.g., idle resources); $b_1$ measures the unit penalty for insufficient supply; $a_2$ measures the unit penalty of being unable to meet the demand; $b_2$ measures the unit overage cost; and the nonnegative constant $k$ serves to rescale the transportation cost, reflecting the relative importance between transport and mismatch costs. These parameters are application-dependent and can be pre-defined accordingly. For instance, in disaster-relief logistics, where delivery costs are secondary to meeting urgent demand, a small $k$ emphasizes mismatch penalties; In ride-sourcing applications, where both costs are typically measured in monetary units, a natural choice is $k=1$.
>
> On the theoretical side, our framework is general and flexible, and the full analysis applies across the entire range of model parameters. In particular, classical optimal transport under balanced settings emerges as a special case within our formulation. The optimization algorithm we present is tailored to a special semi-discrete case, and it is included as proof-of-concept application of our theoretical findings rather than a central contribution of the paper. Notably, it involves minimal hyperparameter tuning: thanks to our analytical derivations, explicit gradients are available, allowing for direct use of standard gradient descent methods. In practice, step size is the only parameter that typically requires adjustment, an issue that is well-understood and extensively studied in the optimization literature. We appreciate the reviewer’s suggestion to elaborate further on implementation details, but we have opted not to include them in the main manuscript for two reasons: (1) to keep the focus on the core theoretical contributions, and (2) to stay within the page limit constraints. We hope this clarifies our design choices and appreciate your understanding.
>
>
> Thank you once again for your insightful and constructive feedback on our manuscript. We hope you find our revision satisfactory.

---

> > ### Comment · Reviewer_z3hw · 2025-08-05
> >
> > Thank you for your comprehensive response to my comments. At this stage, I will retain my current score. I hope that the revised version will provide a thorough clarification of the concerns outlined above.

---

> > > ### Author Response · Authors · 2025-08-06
> > >
> > > Thank you once again for your constructive comments and thoughtful feedback. We will try to further enhance the paper’s exposition by incorporating your valuable suggestions.

---

### Official Review · Reviewer_ZGLs · 2025-07-03

**Clarity:** 3
**Significance:** 3
**Originality:** 3
**Rating:** 4
**Confidence:** 3

**Summary:**

This paper introduces a formulation of unbalanced weak optimal transport incorporating total variation (TV) penalties. Motivated by resource allocation problems with asymmetric over- and under-supply/demand costs, the authors develop a general theoretical framework encompassing existence results and dual formulation. Focusing on the semi-discrete setting—where demand is modeled as a continuous measure and supply as a finite set of discrete locations—the authors develop a tessellation-based reformulation. This yields a dual optimization problem defined over generalized Laguerre cells. They also extend the framework to a quantization problem in which the locations and weights of the discrete supply points are jointly optimized.
The paper presents illustrative examples demonstrating the method’s ability to produce spatially meaningful solutions for simple supply area division and spatial ressource allocation problems.

**Questions:**

- When using a TV penalty in a discrete UOT formulation, it is unclear how mass creation can appear (it is always cheaper to remove mass than adding some mass). Could you give an example for which it happens in your formulation (eq. 1 and 2)?
- How does your method compares to other semi-discrete unbalanced OT methods in the application scenario you have considered?
- How does the setting in section 4.2, where the continuous formulation is sampled, is a semi-discrete problem?


Other comments:
- l123-125: could you provide a reference?
- symbol $^+$ should be defined
- l140: $k$ should be defined
- l233: $d$ should be defined

**Ethical Concerns:**

["NO or VERY MINOR ethics concerns only"]

**Final Justification:**

I have carefully read the authors response and the discussions with other reviewers. While I found some drawbacks to the paper (mainly insufficient motivation — but should be fixable in the camera ready version— and lack of validation in challenging setups different that ressource allocation pbs), I still leaning toward acceptance as it provides sound new developments about weak and unbalanced OT.

**Limitations:**

yes

**Quality:**

3

**Strengths And Weaknesses:**

Strengths:
- The unification of weak OT and unbalanced OT under a TV penalty is original and well-motivated.
- The paper is clearly written, with proofs and intuitive explanations that make the technical contributions accessible.
- The application to spatial allocation and facility location is convincing to some extent

Weaknesses:
- experimental validation: while the examples are illustrative, the empirical evaluation could be expanded. For instance, comparisons with existing entropy-regularized unbalanced OT or standard quantization approaches would strengthen the claims of practical benefit. In addition, the computational complexity of the optimization procedures, especially in higher dimensions or with large discrete supports, could be more thoroughly analyzed
- motivation: while the use of a semi-discrete formulation with mass relaxation makes sense in the application that are described in the spatial ressource allocation problem, the motivation and advantage of instroducing weak OT into the formulation is unsufficiently discussed.
- some formulations and results differ from the standard UOT with TV penalty setting, which is not discussed (see questions below)

---

> ### Author Rebuttal · Authors · 2025-07-31
>
> We sincerely appreciate your positive feedback on our work, as well as your thoughtful and constructive comments. Please find our detailed responses to each of your questions below.
>
> **[1]** When using a TV penalty in a discrete UOT formulation, it is unclear how mass creation can appear (it is always cheaper to remove mass than adding some mass). Could you give an example for which it happens in your formulation (eq. 1 and 2)?
>
> ## R1
> We are glad to answer this question. In the unbalanced setting we consider, the two marginal distributions, $\mu_1$ (supply) and $\mu_2$ (demand), are fixed and determined by the specific application scenario. When the total available supply $|\mu_1|$ is strictly less than the total demand $|\mu_2|$, simply redistributing the existing mass may not be sufficient to meet the demand. In such cases, mass must be created in order to satisfy $\mu_2$, which, in real-world terms, corresponds to a supply shortage. To account for this, we introduce a penalty term that quantifies the cost of mass imbalance, allowing the transport plan to create or delete mass where needed. This formulation enables us to minimize the combined transport and mismatch cost in a principled way. On one hand, the resulting transport plan offers insight into how much additional supply would need to be supplemented, or how much existing supply might be in excess. On the other hand, our framework naturally encompasses the classical balanced setting as a special case, in which exact matching between supply and demand is enforced on both sides.
>
> A simple one-dimensional example helps to illustrate this point. Let $X=[0,1]$, with a continuous supply distribution $\mu_1(dx) =0.3dx$ (total mass 0.3), and a discrete demand $\mu_2=0.5\delta_{0.5}$ (total mass 0.5). Suppose the transport cost is zero, and let the mismatch penalties satisfy $a_1=b_1=0, a_2, b_2>0$. Then the transport plan $\gamma(dx,dy)=0.5dx\ \delta_{0.5}(dy)$ ships all available mass to $y=0.5$ and creates an additional $0.2$ units of mass to fully meet the demand. This plan is in fact optimal, as the OT loss is zero (transportation cost is zero, no penalty on the supply side since $a_1=b_1=0$, and the mismatch penalty on the demand side is also zero).
>
> This example demonstrates that the model flexibly allows for mass creation, mass deletion, and transport, depending on the cost structure and penalty parameters. Such flexibility is critical for capturing real-world resource allocation problems where perfect balance between supply and demand is often unattainable.
>
>
> **[2]** How does your method compare to other semi-discrete unbalanced OT methods in the application scenario you have considered?
>
> ## R2
> Thank you for this insightful question. To the best of our knowledge, there is currently no existing work on semi-discrete unbalanced OT that optimizes a total variation (TV)-regularized objective. The algorithms cited in the literature cannot be adapted by simply replacing the divergence function, as their convergence guarantees critically depend on the strong convexity and smoothness properties of the entropic regularization.
>
> In contrast, we adopt TV regularization, motivated not only by theoretical interest but also by practical relevance. TV loss is intuitive, interpretable, and directly proportional to the number of unsatisfied or excessive units. For example, DiDi, a major ride-sourcing platform in China, has used TV-based metrics to evaluate loss resulting from idle supply or unmet demand (see [9]). While entropy-based regularization (commonly used for computational reasons, see [1,2,3,4,8]) is widespread, it is often less interpretable in practice. In this regard, TV-based regularization remains underexplored in the theoretical OT literature, despite its operational appeal. We believe our incorporation of unbalanced marginal constraints with TV penalties offers a natural and meaningful modeling alternative.
>
> In fact, the main contribution of our paper lies in the development of a comprehensive theoretical framework for a new class of unbalanced weak OT problems incorporating TV penalties, together with an illustration of its applicability to spatial resource allocation. Building on the foundational work in [8], our study takes a significant step forward. To the best of our knowledge, this is the first theoretical treatment of weak OT problems with TV penalties in the unbalanced setting. From a methodological standpoint, the techniques developed in prior works based on entropic regularization are not directly applicable to our setting. This is because the TV penalty exhibits linear growth, in contrast to the smooth and strongly convex structure induced by entropy. As a result, we develop new theoretical tools and analysis techniques to address the challenges posed by the TV-regularized formulation and establish the corresponding duality and optimality conditions in parallel.
>
> **[3]** How is the setting in section 4.2, where the continuous formulation is sampled, a semi-discrete problem?
>
> ## R3
> Thank you for your careful reading and thoughtful question. A semi-discrete optimal transport problem refers to a setting where one marginal measure is absolutely continuous, and the other is discrete. The sampling step we introduce does not alter the mathematical nature of the problem, it simply replaces the continuous integral in the outer objective $$Q(y)=\sum_{i}\int_{C_i(w^{\star})} z(x,i)\,d\mu(x)+a_1\mu(R),$$ with an unbiased empirical estimate, enabling more efficient gradient evaluation. We adopt this sampling-based approximation for computational convenience, particularly since the number of samples is large enough to ensure high accuracy in practice.
>
> We acknowledge that, in a strictly formal sense, this empirical approximation introduces some difference from the idealized formulation. However, based on our numerical experiments, we find that the approximation is highly accurate, and the resulting ot solutions are very close to those obtained via exact integration. Thus, the stochastic estimate merely facilitates efficient gradient computation, allowing the outer–inner optimization framework (Theorem 3.5 & Proposition 3.2) to converge effectively in practice.
>
> **[4]** Other comments:（1）l123-125: could you provide a reference?（2）symbol $^+$ should be defined（3）l140: $k$ should be defined（4）l233: $d$ should be defined
>
> ## R4
> We thank the reviewer for the careful reading and detailed, point-by-point comments. We will revise the manuscript accordingly. Below are our responses and clarifications:
>
> (1) The newsvendor loss function is commonly used in operations management and economics to measure the supply-demand mismatch (see, e.g., [5,6,7]). This form of penalty has also been adopted in practice, e.g., by DiDi, a major ride-sourcing platform (see [9]). Specifically, it penalizes the mismatches from both sides: If supply exceeds demand, a proportional overage cost is incurred, reflecting the cost of excess supply (e.g., idle resources, inventory holding); When demand exceeds supply, a shortage cost is applied, corresponding to the opportunity loss from unmet demand, such as forgone revenue. This structure allows the penalty to directly reflect economic losses in a way that is both intuitive and interpretable.
>
> (2) For any signed object, the superscript “+” denotes its positive part. Specifically, for a scalar or measurable function $g$ we set $g^+=\max\{g,0\}$, while for a signed measure $\nu$ on $(X,\mathcal{F})$, we take $\nu^+$ to be the positive variation in the Jordan decomposition $\nu=\nu^++\nu^-$ with $\nu^+\ge0$, $\nu^-\ge 0$ and $\nu^+ \perp\nu^-$.
>
> (3) The nonnegative constant $k$ serves to rescale the transportation cost, reflecting the relative importance between transport and mismatch costs. While technically it can be absorbed into the definition of the cost function $c$, keeping it explicit highlights the application-dependent trade-off. For instance, in disaster-relief logistics, where delivery costs are secondary to meeting urgent demand, a small $k$ emphasizes mismatch penalties; In ride-sourcing applications, where both costs are typically measured in monetary units, a natural choice is $k=1$.
>
> (4) d(x,y) is the Euclidean distance between $x$ and $y$.
>
> **[5]** Regarding the clarification on our motivations, we kindly ask you to refer to our response to Reviewer KBQN (R2), as we are constrained by the character limit imposed by the conference system. We greatly appreciate your understanding.
>
> Thank you once again for your insightful and constructive feedback on our manuscript. We hope you find our revision satisfactory.
>
>
> [1] Chizat, Lenaic, et al. Scaling algorithms for unbalanced optimal transport problems. Mathematics of computation 87.314 (2018): 2563-2609.
>
> [2] Scetbon, Meyer, et al. Unbalanced low-rank optimal transport solvers. NeurIPS 36 (2023): 52312-52325.
>
> [3] Pooladian, Aram-Alexandre, Vincent Divol, and Jonathan Niles-Weed. Minimax estimation of discontinuous optimal transport maps: The semi-discrete case. ICML, 2023.
>
> [4] Bonet, Clément, et al. Slicing Unbalanced Optimal Transport. TMLR (2024).
>
> [5] Petruzzi, Nicholas C., and Maqbool Dada. Pricing and the newsvendor problem: A review with extensions. Operations Research 47.2 (1999): 183-194.
>
> [6] Qin, Yan, et al. The newsvendor problem: Review and directions for future research. EJOR 213.2 (2011): 361-374.
>
> [7] Khouja, Moutaz. The single-period (news-vendor) problem: literature review and suggestions for future research. omega 27.5 (1999): 537-553.
>
> [8] Matthias Liero, Alexander Mielke, and Giuseppe Savaré. Optimal entropy-transport problems and a new Hellinger–Kantorovich distance between positive measures. Inventiones Mathematicae, 211(3):969–1117, 2018.
>
> [9] Fan Zhou, Shikai Luo, Xiaohu Qie, Jieping Ye, and Hongtu Zhu. Graph-based equilibrium metrics for dynamic supply–demand systems with applications to ride-sourcing platforms. JASA, 116(536):1688–1699, 2021.

---

> > ### Author Response · Authors · 2025-08-07
> > **Kind Reminder: Response to Rebuttal from NeurIPS Submission 12974**
> >
> > Dear Reviewer ZGLs,
> >
> > Thank you very much for your insightful and positive comments on our submission. We are also pleased to see that the review team appreciates our work and has provided supportive feedback overall.
> >
> > As the discussion phase comes to a close, we would like to kindly ask whether our rebuttal has fully addressed your concerns, and whether this may have influenced your assessment of our manuscript.
> >
> > In our previous rebuttal, we provided detailed clarifications to each of your questions. We will also try to incorporate those clarifications into the revised manuscript, as we believe they will further improve the paper. If there are any remaining issues or further questions, we would be more than happy to discuss them. Should you feel that our responses have resolved your concerns, we would be grateful for your consideration of an updated evaluation.
> >
> > Thanks again for your time and constructive input throughout this process. We look forward to your final thoughts.
> >
> > Best regards,
> > The Authors

---

> > > ### Comment · Reviewer_ZGLs · 2025-08-09
> > >
> > > Thank you very much for your detailed answer. I keep my positive evaluation of the paper.

---

> > > > ### Author Response · Authors · 2025-08-09
> > > > **Follow-up on Rebuttal and Appreciation for Your Kind Comments**
> > > >
> > > > Dear Reviewer ZGLs,
> > > >
> > > > We sincerely appreciate the time you have taken to read our rebuttal and for your positive evaluation of our work. We are very glad to know that our responses addressed your concerns.
> > > >
> > > > If there are no remaining issues, we would be grateful if you might consider raising your score in light of the additional clarifications and results we have provided.
> > > >
> > > > Thank you again for your thoughtful review and for helping ensure a constructive and fair evaluation process.
> > > >
> > > > Best regards,
> > > > The Authors

---

### Official Review · Reviewer_ZhcG · 2025-07-03

**Clarity:** 4
**Significance:** 2
**Originality:** 3
**Rating:** 5
**Confidence:** 3

**Summary:**

The paper presents a new family of unbalanced weak optimal transport with weighted total variation penalties with a focus on the semi-discrete setting. They present a comprehensive theoretical analysis that yields (1) the dual formulation for the general problem, and in the special semi-discrete cases (2) a finite-dimensional tessellation reformulation (corresponding to service area partition) that can be solved via gradient methods, and (3) a sequential optimization strategy for optimal spatial resource allocation. Illustrative numerical examples well showcase the motivations of the OT problem as well as its numerical tractability.

**Questions:**

My questions are mostly related to numerical convergence.
1. You set a decaying step size in Section 4, which is pretty understandable for the piecewise nonsmooth objective. However, I am concerned that if you scale up the problem size, such decaying step design may converge only suboptimally.  Did you try different problem sizes? How bad is the condition number of the problem as we scale it up?
2. While the sequential or alternating method looks viable in Sec 3.3, no convergence analysis is provided. I don’t see this finite-dimensional problem still preserving convexity with respect to both locations and masses. Can you comment on this a bit? I would imagine its performance resembles the EM algorithm for gaussian mixtures.

The last question is a bit more general.

3. Can you comment on the significance of recasting these familiar problems including resource allocation commonly appears in operations research literature into an optimal transport formulation? What are the key pros and cons compared to a classical linear program approach?

**Ethical Concerns:**

["NO or VERY MINOR ethics concerns only"]

**Final Justification:**

I generally recommend the acceptance for their clear motivations, complete theoretical framework, as well as certain level of guidance on algorithm design. I think the paper also writes clearly and is thus suitable for circulation of ideas in the current venue.

On the other hand, I am not sufficiently sure of the theoretical novelty. The empirical evidence and algorithmic contribution is also not too strong as noted by most reviewers, yet I am fine with this.

**Limitations:**

Yes, the authors acknowledged insufficiency of additional applications and discussed future work in the last two sections.

**Quality:**

3

**Strengths And Weaknesses:**

**Strengths**
1. The paper is easy to follow. It is very well motivated and self-contained, with a clear structure, mathematical rigor, and accompanied by intuitive interpretations of abstract concepts.
2. The proposed OT problem looks important and can serve a unified framework for several important real-world problems.
3. The theory is comprehensive and elegant with clear citations showing connections with related literature. They look correct, though detailed proofs are not checked with confidence as the reviewer has insufficient knowledge of deep theories. Provided gradient derivations are helpful to develop quick algorithms.

**Weaknesses**
1. The experiments are relatively toy level, though I agree that the paper’s focus is more theoretical. It would be interesting to see how scaling up the problem size affects the efficiency of the presented gradient methods.
2. The applications considered are relatively restricted. The example presented in Example 2.1 is not too different from the field of the motivating resource allocation problems. It would definitely be interesting to see if applications in other fields may be casted to the considered OT problem.

---

> ### Author Rebuttal · Authors · 2025-07-31
>
> We sincerely appreciate your positive feedback on our work, as well as your thoughtful and constructive comments. Please find our detailed responses to each of your questions below.
>
> **[1]** I am concerned that if you scale up the problem size, such a decaying step design may converge only suboptimally. Did you try different problem sizes? How bad is the condition number of the problem as we scale it up?
> ## R1
> Thanks for your insightful comments. We fully agree that for less smooth objectives, gradient descent methods may face challenges in performance and convergence to the optimal solution. Nevertheless, our derivation of explicit gradient expressions enables us to experiment with gradient-based methods, and we appreciate your understanding that this section serves primarily as an illustrative application of our theoretical findings. We believe the application domain, including the one you mentioned, contains many interesting directions that warrant deeper exploration in future work.
>
> In response to your suggestion, we have conducted preliminary experiments to assess scalability and examine the impact of the condition number. We apologize that these experiments remain preliminary due to time constraints and page limit, but we are committed to pursuing a more thorough investigation in future research.
>
> Specifically, in terms of step size, we note that the decaying rate and initial value can be easily adjusted if convergence appears insufficient. As an initial scalability study, we repeated the depot-location experiment from Section 4.2 at five progressively larger scales, increasing the dimensionality of the optimization variable. We sample locations from a two‑component Gaussian mixture with means $(0.3d,0.3d)$ and $(0.7d,0.7d)$. The number of depots is $m$ and the decaying step size of $0.3/(1+t)^{0.6}$ is used.
>
> The table below shows that while the empirical condition number of the smoothed Hessian does increase from approximately $9$ to over $7000$ as the number of optimization parameters grows from $12$ to $192$. Despite this, the CPU time scaled roughly linearly, remaining within a practically acceptable range. While these results are not central to our main contributions, we believe they suggest that our theoretical findings could scale reasonably well and support potential use for larger-scale problem instances. We thank the reviewer again for prompting this analysis and look forward to exploring these aspects in more detail in future work.
> | scale (m,d) | (4,1) | (8,2) |(16,3) | (32,3) |(64,4)|
> |-|-|-|-|-|-|
> | numbers of parameter | 12 | 24 | 48 | 96 |192|
> | condition number | 8.8 | 9.5 | 117.7| 885.6 |7437|
> |CPU time |4.9s|7.5s|15.2s|29.6s|65.9s|
>
>
> **[2]** While the sequential or alternating method looks viable in Sec 3.3, no convergence analysis is provided. I don’t see this finite-dimensional problem still preserving convexity to both locations and masses. Can you comment on this a bit? I would imagine its performance resembles the EM algorithm for Gaussian mixtures.
> ## R2
> We sincerely thank the reviewer for the insightful and professional remark. You are absolutely right in observing that, since the outer objective is not jointly convex in $(Y,w)$, our alternating scheme (similar in spirit to the EM algorithm for Gaussian mixtures) may only have convergence to a stationary point, which may be a local rather than global optimum. This limitation is indeed inherent to first-order methods in non-convex settings. Nonetheless, the explicit gradient expressions we derive offer significant computational convenience and enable practical implementation of the algorithm.
>
> While a full convergence analysis lies outside the scope of our main theoretical contribution, we believe that the availability of explicit gradient expressions makes it relatively straightforward to extend standard convergence guarantees, such as convergence to stationary points under mild assumptions, to our setting. Of course, in the special case where the objective is convex, convergence to the unique global optimum would follow. However, this is not the case for the general formulation considered in our work. In general, when we apply gradient descent, the algorithm can converge to the stationary set of the ODE constructed by the gradient flow, see [1,2] for details. Furthermore, when some respective convexity conditions hold, the method fits the gradient block‑coordinate descent setting studied by [3,4]. Specifically, alternating updating methods can also achieve convergence because we have explicit gradients, and the analysis is quite standard [5,6], so we omit it here.
>
> In summary, although global optimality cannot be guaranteed due to non-convexity, our algorithm provides a tractable and theoretically grounded method for identifying stationary solutions, which can serve as useful approximations in practical spatial resource allocation tasks.
>
> **[3]** Can you comment on the significance of recasting these familiar problems including resource allocation commonly appears in operations research literature into an optimal transport formulation? What are the key pros and cons compared to a classical linear program approach?
> ## R3
> We thank the reviewer for this thought-provoking question. Recasting classical resource allocation problems as an unbalanced, weak optimal transport (OT) problem offers several distinct advantages from both theoretical and practical perspectives. We elaborate on the key benefits below:
>
> (1) Geometric expressiveness. OT is inherently grounded in a metric structure, allowing the concept of “distance” in spatial resource allocation to be represented and leveraged more naturally. In particular, the semi-discrete OT setting enables the use of Laguerre/Voronoi tessellations and other geometry-aware algorithms that would be difficult to derive from purely algebraic linear programming (LP) formulations. Importantly, this geometric perspective is not only more theoretically elegant but also enhances practical decision-making, e.g., in partitioning service areas for facilities like fireboat stations or inventory warehouses.
>
> (2) Natural treatment of imbalance. Through the inclusion of a total variation (TV) penalty, the unbalanced OT formulation directly models the cost of mass creation or deletion. This provides an explicit and tunable trade-off between transport cost and supply-demand mismatch, expressed in the same units. In contrast, LP models typically rely on artificial slack variables and large-penalty terms. The magnitude of the TV penalty offers a clear and quantitative measure of over- or under-supply severity.
>
> (3) Semi‑discrete realism. Many spatial allocation problems involve a continuous customer distribution and a finite set of candidate facilities, a structure that the semi-discrete OT framework captures precisely. This allows one to integrate over the continuous domain (analytically or via Monte Carlo methods), while keeping the facility variables finite-dimensional and interpretable. While in practice customers may be discrete, modeling them as a density field enables cleaner structural insights and facilitates scalable approximations without sacrificing realism.
>
> (4) Scalability and Structural Efficiency. A standard approach in continuous domains is to discretize space and solve the resulting high-dimensional discrete OT problem via LP. However, this quickly becomes computationally infeasible as the dimension grows or finer resolution is desired. By contrast, the OT formulation, particularly in the semi-discrete case, allows us to reduce the problem to a low-dimensional weight vector, with the same size as the number of facilities. This drastically improves scalability compared to large LPs, especially in high dimensions.
>
> Together, these features make the OT-based framework a geometrically faithful, imbalance-aware, and practically aligned modeling language for spatial resource allocation, offering a clear advantage over traditional LP formulations.
>
> That said, we acknowledge that the OT approach is not without limitations. LP models, for instance, offer immediate implementability and require little structural analysis; they can yield a solution (good or bad) with minimal modeling and theoretical analysis effort, especially after discretization. Furthermore, beyond the semi-discrete setting, the theoretical results in the general OT framework may be more abstract and harder to interpret for direct application. Nevertheless, we believe the enhanced structure, interpretability, and flexibility afforded by the OT perspective provide compelling benefits in many settings.
>
>
> Finally, we acknowledge the insufficiency of additional applications because we want to focus more on our theoretical framework in this theory-track paper. In future work, we will follow your inspiring suggestions to explore more applications based on the foundation we have established; indeed, some of the potential extensions or applications have been discussed in the last two sections.
>
>
> Thank you once again for your insightful and constructive feedback on our manuscript. We hope you find our revision satisfactory.
>
> [1] Borkar, Vivek S., and Vivek S. Borkar. Stochastic approximation: a dynamical systems viewpoint. Vol. 100. Cambridge: Cambridge University Press, 2008.
>
> [2] Kushner, Harold J., and G. George Yin. Stochastic approximation and recursive algorithms and applications. New York, NY: Springer New York, 2003.
>
> [3] Tseng, Paul. "Convergence of a block coordinate descent method for nondifferentiable minimization." JOTA 109.3 (2001): 475-494.
>
> [4] Beck, Amir, and Luba Tetruashvili. "On the convergence of block coordinate descent type methods." SIOpt 23.4 (2013): 2037-2060.
>
> [5] Wright, Stephen J. "Coordinate descent algorithms." MP 151.1 (2015): 3-34.
>
> [6] Lin, Tianyi, Chi Jin, and Michael I. Jordan. "Two-timescale gradient descent ascent algorithms for nonconvex minimax optimization." JMLR 26.11 (2025): 1-45.

---

> > ### Comment · Reviewer_ZhcG · 2025-08-02
> >
> > Thank you for your detailed responses and I find the responses satisfying. I believe it would be helpful to briefly incorporate those bullets in your R3 into revision for help intuitive and quick appreciation of your proposal. While I have no outstanding questions, I will keep monitor your discussions with other reviewers.

---

> > > ### Author Response · Authors · 2025-08-06
> > >
> > > Thank you once again for your constructive comments and kind words of appreciation. We are pleased to know that you are satisfied with our responses, and we will try to incorporate those clarifications into the revised manuscript.

---

### Official Review · Reviewer_KBQN · 2025-07-05

**Clarity:** 2
**Significance:** 2
**Originality:** 3
**Rating:** 3
**Confidence:** 3

**Summary:**

The paper develops a semi-discrete formulation of weak optimal transport where the discrepancies between marginal distributions are regularized through a TV term.  And this work proves the existence of an optimal plan and gives a dual characterisation whose variables coincide with the weights that generate Voronoi diagram. They proposed an iterative weight–update algorithm for the case of a continuous demand distribution and finitely many depots.  They applied their methods on two small two dimensional demos to shows how the method allocates spatial resources.

**Questions:**

Could you position your algorithm more explicitly against the approaches of Agarwal et al.?  Do you have either theoretical bounds or empirical results on the convergence rate and scalability of the weight–update scheme?  Why was total variation selected over other divergences commonly used in OT?  Would you clarify the distinction you intend between weak and unbalanced OT, and explain whether your formulation is essentially the standard one–sided unbalanced model?

**Ethical Concerns:**

["NO or VERY MINOR ethics concerns only"]

**Final Justification:**

I have reviewed and acknowledged the the authors' response. I think the authors made responses to each of the concerns. After reviewed the discussion with the other reviewers I will keep my initial justification. Thanks!

**Quality:**

3

**Strengths And Weaknesses:**

The theoretical proof is good, optimal existence and duality are established, the notation is clear and consistence, and the link between the dual weights and Laguerre tessellations is well explained.  But the bridge to Laguerre diagrams is not novel, having been analysed in detail in recent work about semi-OT by Agarwal et al. (SODA 2024, NeurIPS 2024), so the main incremental value appears to be the introduction of a total\-variation penalty within the semi-OT framework. The manuscript lists four extensions—weak transport cost, unbalanced marginal constraints, total\-variation regularisation and semi\-discrete geometry—but their joint motivation is not so clear, kind confusing why all four are needed together. And, the terminology "weak OT" and "one–sided unbalanced OT" are sometimes conflates, which may confuse readers. Empirical support is limited to small toy instances, with no complexity analysis, convergence guarantee or sensitivity study.

- Agarwal, Pankaj K., et al. "Fast and accurate approximations of the optimal transport in semi-discrete and discrete settings." Proceedings of the 2024 Annual ACM-SIAM Symposium on Discrete Algorithms (SODA). Society for Industrial and Applied Mathematics, 2024.
- Agarwal, Pankaj, et al. "A combinatorial algorithm for the semi-discrete optimal transport problem." Advances in Neural Information Processing Systems 37 (2024): 25857-25887.

---

> ### Author Rebuttal · Authors · 2025-07-31
>
> We have carefully reviewed and summarized your comments and questions into five main points below. While they may not appear in the exact order presented, we have addressed each of them in turn. Thanks for your thoughtful and constructive feedback.
>
> **[1]** The bridge to Laguerre diagrams is not novel, having been analyzed in recent work about semi-OT. Could you position your algorithm more explicitly against the approaches of Agarwal et al.?
>
> ## R1
> Thank you for this insightful comment. We fully agree that the use of Laguerre diagrams in semi-discrete OT problems is not our novelty, but rather a natural tool in analyzing such problems (see e.g., [1]). The main contribution of our paper lies in the development of a comprehensive theoretical framework for a new class of unbalanced weak OT problems incorporating total variation (TV) penalties, along with an illustration of its applicability to spatial resource allocation. Building on the work [4], we take one step forward, and to the best of our knowledge, this is the first theoretical study to address weak OT problems with TV penalties in the unbalanced setting. Importantly, the introduction of TV penalties is not only technically motivated by the need to handle mismatched masses, but also practically justified as real-world problems often involve supply-demand imbalance, making such a formulation both reasonable and necessary (see also [5]).
>
> We are glad to clarify our contribution with the recent work of Agarwal et al. (SODA’24, NeurIPS’24). While their studies focus on the balanced semi-discrete OT setting, where the source $\mu$ and target $\nu$ are probability measures with equal total mass, our work advances the framework in two key directions. First, we consider a more general unbalanced setting in which the source and target measures may have unequal total mass. This naturally introduces a mismatch, for which we introduce TV penalties to regularize the deviation. This extension is not only mathematically nontrivial but also grounded in practical scenarios where supply rarely matches demand exactly. Second, within this generalized model, we develop a complete duality theory that lays the foundation for further analysis and a range of potential applications. For example, we derive explicit sub-gradients for the dual variables, enabling the construction of transport plans through simple gradient-based algorithms. Additionally, we propose a method that jointly optimizes both the cell weights and facility locations.
>
> In summary, as a theory track paper, we believe our submission offers a substantial theoretical contribution that generalizes classical results to the unbalanced and practically motivated setting, supported by a rigorous dual framework. We hope these clarifications help better position our work within the growing body of research on OT.
>
> **[2]** The manuscript lists four extensions—weak OT, unbalanced marginal constraints, total-variation regularization, and semi-discrete geometry—but their joint motivation is not so clear.
>
> ## R2
> Thanks for your comment, and we would like to make the following clarifications.
>
> (1) As replied in R1, we consider the unbalanced OT setting, which is motivated by many practical scenarios involving mismatched supply and demand, e.g., the aggregate number of available taxis may exceed the number of potential customers. In such cases, introducing regularization is essential to penalize deviations on either the source (supply) or target (demand) side; otherwise, the problem may not be well-posed. We adopt total-variation regularization which is also motivated by practical relevance: TV loss is intuitive and proportional to the count of unsatisfied or excessive units. In contrast to entropy regularization (commonly used for numerical convenience, see [4]), TV regularization is more interpretable and operationally grounded, yet remains underexplored in the theoretical OT literature. Thus, our incorporation of unbalanced marginal constraints and TV penalties is both natural and meaningful.
>
> (2) We indeed solve the general OT problem dealing with general source $\mu$ and target $\nu$ in the main part (see Section 3.1). We subsequently specialize to the semi-discrete case in Section 3.2, primarily due to its favorable and interpretable geometric structure, which facilitates further analysis and insights (see Section 3.2). This semi-discrete geometry is not simply an extension, but rather a natural and illustrative special case within our broader framework that warrants deeper investigation.
>
> (3) Finally, from a theoretical standpoint, our goal is to develop as general a framework as possible. The weak formulation enables us to push the theory to its limits and includes classical OT as a special case. Beyond its generality, the weak formulation has shown promising practical advantages: by relaxing the coupling constraints, it enhances robustness to sampling noise and stability in empirical settings where exact couplings may not exist or may be sensitive to perturbations (see, e.g., [2,3]).
>
> **[3]** The terminology “weak OT” and “one–sided unbalanced OT” are sometimes conflated. Would you clarify the distinction you intend between weak and unbalanced OT, and explain whether your formulation is essentially the standard one–sided unbalanced model?
>
> ## R3
> We appreciate the chance to clarify the terminology. Weak OT and unbalanced OT address two distinct and orthogonal modeling dimensions. Weak OT modifies how transport cost is evaluated.  Rather than paying for every point-to-point flow, it allows each source point to distribute its mass as a probability measure over the target space. The cost is then computed based on the barycenter of this distribution. This formulation preserves total mass but introduces stochastic routing, which can enhance robustness and flexibility. In contrast, unbalanced OT retains the classical point-to-point transport cost but relaxes marginal constraints, permitting mass surplus or deficit on one or both sides. Such discrepancies are penalized via total variation (TV) in our paper. For example, in a ride-hailing scenario, the platform may only control the supply (taxis), not the demand (customers). This leads to a one-sided, unbalanced setting, where the supply distribution is fixed, and deviations in meeting demand are penalized. Since these two generalizations pertain to different aspects: the former to cost structure and the latter to marginal feasibility, they are complementary to model practical situations and can be combined, resulting in what we refer to as “weak and unbalanced OT.” We hope this distinction helps clarify the modeling choices in our framework.
>
> **[4]** Do you have either theoretical bounds or empirical results on the convergence rate and scalability of the weight–update scheme?
>
> ## R4
> Thanks for your comment. As a theory-track submission, the primary contribution of our work lies in the development of a comprehensive theoretical framework for a new class of unbalanced weak OT problems incorporating total variation (TV) penalties. The optimization routine presented in the (special) semi-discrete setting is therefore intended not as the main focus, but rather as an illustrative application of the proposed framework. That said, we appreciate your suggestion to incorporate complexity analysis, convergence guarantees, and sensitivity studies. While these are certainly valuable directions, we have opted to leave them for future work for two reasons: (i) to maintain clarity and focus on the general theoretical framework, and (ii) due to space limitations. Nonetheless, we note that the use of explicit gradients in our setting should allow for relatively straightforward extensions of standard convergence analyses in the semi-discrete case.
>
> We also appreciate your concern regarding scalability. As an illustration, we extend the depot-location experiment in Section 4.2 to five larger scales, increasing the dimensionality of the optimization variable. The CPU time is acceptable. Although this scalability assessment is not the focus of our paper, we believe it demonstrates the practical potential of our theoretical findings and provides a promising direction for future investigation. Since we are unable to upload images or PDF files, we present the results in tabular form.
> | scale (m,d) | (4,1) | (8,2) |(16,3) | (32,3) |(64,4)|
> |-|-|-|-|-|-|
> | numbers of parameter | 12 | 24 | 48 | 96 |192|
> |CPU time |4.9s|7.5s|15.2s|29.6s|65.9s|
>
>
> **[5]** Why was TV selected over other divergences commonly used in OT?
>
> ## R5
> Thanks for raising this important point. As replied in R2, we chose the TV penalty primarily because it offers a clear and intuitive economic interpretation, closely aligned with the well-known newsvendor loss function commonly used in operations management and economics. This form of penalty is also adopted in practice, e.g., by DiDi, a major ride-sourcing platform (see [5]). Specifically, it penalizes the mismatches from both sides: If supply exceeds demand, a proportional overage cost is incurred, reflecting the cost of excess supply (e.g., idle resources, inventory holding); When demand exceeds supply, a shortage cost is applied, corresponding to the opportunity loss from unmet demand, such as forgone revenue. This structure allows the penalty to directly reflect economic losses in a way that is both intuitive and interpretable. Given our focus on practical resource allocation problems as highlighted in the paper’s title, we intentionally adopted the TV regularization instead of entropy-based penalties, which are less straightforward to interpret in economic terms.
>
> [1] Aurenhammer, Klein, Lee. ... World Scientific Publishing Company, 2013.
>
> [2] Backhoff-Veraguas and Pammer. …. Bernoulli, 2022.
>
> [3] Feydy, Séjourné, Vialard, Amari, Trouvé, Peyré. ... AISTATS, 2019.
>
> [4] Liero, Mielke, Savaré. ... Inventiones Mathematicae, 2018.
>
> [5] Zhou, Luo, Qie, Ye, Zhu. ... JASA, 2021.

---

> > ### Author Response · Authors · 2025-08-07
> > **Kind Reminder: Response to Rebuttal from NeurIPS Submission 12974**
> >
> > Dear Reviewer KBQN,
> >
> > Thank you very much for your insightful comments on our submission. We are pleased to see that the review team appreciates our work and has provided positive and supportive feedback overall.
> >
> > As the discussion phase draws to a close, we would like to kindly ask whether our rebuttal has fully addressed your concerns, and whether this may have influenced your assessment of our manuscript.
> >
> > In our previous rebuttal, we provided detailed clarifications to each of your questions. If there are any remaining issues or further questions, we would be more than happy to discuss them. If our clarifications have been helpful, and you feel an updated evaluation is appropriate, we would be grateful for your consideration.
> >
> > Thanks again for your time and constructive input throughout this process. We look forward to your final thoughts.
> >
> > Best regards,
> > The Authors

---

### Decision · Program_Chairs · 2025-09-17

**Decision:**

Accept (poster)

**Comment:**

This paper introduces and analyzes a new class of unbalanced weak optimal transport problems with total variation penalties, motivated by spatial resource allocation tasks such as facility placement or service area partitioning. All reviewers agree that this is a strong paper and should be accepted.

The reviewers highlighted that the paper makes interesting and valuable contributions. The unification of weak OT and unbalanced OT under a TV penalty is both novel and well-motivated, and the authors provide a thorough theoretical treatment. The tessellation-based structure is elegant and leads to explicit gradient expressions that support practical algorithmic developments. The paper is clearly written, with careful proofs and intuitive explanations, and the application to spatial allocation and facility location problems adds further relevance and clarity.

Some weaknesses were also noted. The experiments, while illustrative, are limited to small-scale examples and do not include comparisons to alternative methods, and the computational complexity of the optimization procedures is not fully explored. Moreover, although the spatial allocation motivation is clear, the specific role and advantages of introducing weak OT into the formulation could be elaborated further, as could connections to related applied work. The authors provided useful clarifications on these points in their rebuttal and committed to addressing them in the final version, which should strengthen the paper further.

In summary, this is a well-written and rigorous work that introduces a novel and well-founded framework with solid theoretical contributions and meaningful practical motivation. I recommend acceptance.